# CHAIN-OF-INFLUENCE: TRACING INTERDEPENDENCIES ACROSS TIME AND FEATURES IN CLINICAL PREDICTIVE MODELINGS

## ABSTRACT

Modeling clinical time-series data is hampered by the challenge of capturing latent, time-varying dependencies among features. State-of-the-art approaches often rely on black-box mechanisms or simple aggregation, failing to explicitly model how the influence of one clinical variable propagates through others over time. We propose **Chain-of-Influence (CoI)**, an interpretable deep learning framework that constructs an explicit, time-unfolded graph of feature interactions. CoI enables the tracing of influence pathways, providing a granular audit trail that shows how any feature at any time contributes to the final prediction, both directly and through its influence on other variables. We evaluate CoI on mortality and disease progression tasks using the MIMIC-IV dataset and a chronic kidney disease cohort. Our framework achieves state-of-the-art predictive performance (AUROC of 0.960 on CKD progression and 0.950 on ICU mortality), with deletion-based sensitivity analyses confirming that CoI's learned attributions faithfully reflect its decision process. Through case studies, we demonstrate that CoI uncovers clinically meaningful, patient-specific patterns of disease progression, offering enhanced transparency into the temporal and cross-feature dependencies that inform clinical decision-making.

## 1 INTRODUCTION

Chronic Kidney Disease (CKD) affects approximately 8%-16% of the global population and represents one of the most significant challenges in modern healthcare, with its gradual and irreversible progression toward End-Stage Renal Disease (ESRD) imposing substantial clinical and economic burdens National Kidney Foundation (2024); Murphy et al. (2019). The complexity of CKD progression stems from multifaceted interactions between clinical, demographic, and socioeconomic factors that evolve over extended time periods, making accurate prediction and effective management particularly challenging Lin et al. (2013); Li & Padman (2024).

Healthcare data capturing this progression is inherently temporal and multi-dimensional, with clinical outcomes resulting from complex interactions between patient characteristics, laboratory values, medications, and comorbidities. However, accurate prediction alone is insufficient for clinical deployment—clinicians require transparent and interpretable models to confidently incorporate predictive insights into patient care Caruana et al. (2015); Rudin (2019); Tonekaboni et al. (2019). This need for interpretability has driven significant evolution in explainable artificial intelligence (XAI), from traditional post-hoc methods like LIME Ribeiro et al. (2016) and SHAP Lundberg & Lee (2017) toward more sophisticated attention-based approaches that provide intrinsic interpretability Choi et al. (2016; 2017); Bardhan (2024).

Despite these advances, current approaches to clinical prediction face fundamental limitations. Traditional methods either aggregate temporal information into static features, losing crucial temporal dynamics, or model temporal patterns while treating features independently, missing critical inter-feature relationships. Even sophisticated attention-based models like RETAIN Choi et al. (2016), while providing interpretability through temporal and feature-level attention, fail to explicitly model how features influence each other across different time points—a critical gap in understanding disease progression pathways.

These chains of influence—where early clinical indicators propagate through interconnected pathways to affect later outcomes—represent a core challenge for prediction in both chronic and acute care settings. In the slow progression toward End-Stage Renal Disease (ESRD), a patient's declining eGFR over several months may trigger medication adjustments that in turn affect subsequent laboratory values and hospitalization patterns. The timeline is compressed in an Intensive Care Unit (ICU), but the principle is the same: a sudden drop in blood pressure can necessitate a vasopressor, which then impacts heart rate and organ function within hours, directly influencing the immediate risk of mortality. Whether unfolding over years or hours, these complex temporal-feature interdependencies are what current models fail to capture explicitly.

We introduce Chain-of-Influence (CoI), a novel deep learning framework designed to model and interpret these complex temporal-feature interdependencies.

## 2 RELATED WORK

Our work builds upon a rich history of attention-based models designed for clinical time-series data. We categorize prior work into three main areas: foundational attention architectures, models incorporating temporal and structural priors, and transformer-based approaches.

**Foundational Attention Architectures**  Attention mechanisms marked a turning point for interpretability in clinical deep learning. The foundational model in this area, RETAIN, introduced a two-level attention mechanism that operates in reverse chronological order to mimic clinical reasoning Choi et al. (2016). By first weighting the importance of entire patient visits and then individual clinical codes within those visits, RETAIN provides clear, intuitive explanations for its predictions. Following this, models like Dipole extended the concept by using bidirectional LSTMs to capture a more holistic view of the patient's history, exploring various attention formulations to better understand inter-visit dependencies Ma et al. (2017). While these pioneering models successfully identify what clinical events are important and when they occurred, they model these events largely as independent contributors to the final prediction. They do not explicitly capture the influence pathways—how a variable at one point in time may directly affect a different variable later on.

**Time-Aware and Hierarchical Attention**  A significant challenge in clinical data is its inherent irregularity and the varying importance of events over time. To address this, a second wave of research developed more sophisticated temporal attention models. For instance, ATTAIN integrated time-decay factors into its attention weights, allowing the model to dynamically prioritize events based on their temporal proximity to the prediction time Zhang et al. (2019). Other models introduced domain-specific structural priors. StageNet, for example, models disease progression through distinct stages, using a combination of recurrent networks and attention to learn stage-specific representations and their transitions over time Gao et al. (2020). Similarly, AdaCare introduces an adaptive attention mechanism that modulates feature importance based on the patient's evolving disease severity, captured by a separate deep learning component Ma et al. (2020). These approaches refine the temporal and contextual aspects of attention, but the inter-feature dependencies remain encapsulated within the black-box hidden states of recurrent networks, rather than being an explicit and interpretable output of the model.

**Self-Attention and Transformer-Based Approaches**  The Transformer and its self-attention mechanism provide a powerful way to model global, all-to-all interactions within a sequence Vaswani et al. (2017). In the clinical domain, BEHRT treats a patient record as a sequence of concepts and learns contextualized representations Li et al. (2020); PAVE builds on this with multi-layer self-attention to identify important events and aggregate them into higher-level clinical patterns Steinberg et al. (2021); and XTSFormer introduces multi-scale temporal attention to separate short-term acute signals from long-term chronic trends Zhang et al. (2024).

While self-attention captures a dense web of pairwise dependencies, the resulting maps remain difficult to interpret clinically: they highlight correlations but do not clearly expose directed, sequential pathways of influence. In particular, they do not provide a traceable audit trail of how an early event propagates through other variables over time. CoI aims to bridge this gap by explicitly modeling and tracing inter-feature influences across time, moving beyond identifying salient visits or features to quantifying their dynamic interactions.

## 3 METHODOLOGY

In this section, we introduce the Chain-of-Influence (CoI) model, an interpretable deep learning architecture designed to capture and quantify both temporal dynamics and feature interactions in sequential clinical data. The model integrates three complementary attention mechanisms—temporal attention, feature-level attention, and cross-feature attention—into a unified computational graph that enables accurate predictions while providing rich interpretability through explicit influence quantification. Figure 1 provides a complete overview of the model architecture.

### 3.1 MODEL ARCHITECTURE

#### 3.1.1 INPUT REPRESENTATION AND EMBEDDING

Given an input tensor $\mathbf{X} \in \mathbb{R}^{B \times T \times D_{\text{in}}}$, where $B$ denotes batch size, $T$ represents the number of time steps, and $D_{\text{in}}$ is the number of input features, we first project the input into a higher-dimensional embedding space through a learnable linear transformation:

$$\mathbf{E} = \text{Linear}(\mathbf{X}), \quad \mathbf{E} \in \mathbb{R}^{B \times T \times D_{\text{emb}}}, \tag{1}$$

where $D_{\text{emb}}$ denotes the embedding dimension. To encode temporal ordering, we incorporate fixed sinusoidal positional encodings $\text{PE} \in \mathbb{R}^{1 \times T \times D_{\text{emb}}}$, yielding positionally-encoded embeddings:

$$\mathbf{E}_{\text{t}} = \mathbf{E} + \text{PE}, \quad \mathbf{E}_{\text{t}} \in \mathbb{R}^{B \times T \times D_{\text{emb}}}. \tag{2}$$

These embeddings serve as the shared input to three parallel computational pathways described below.

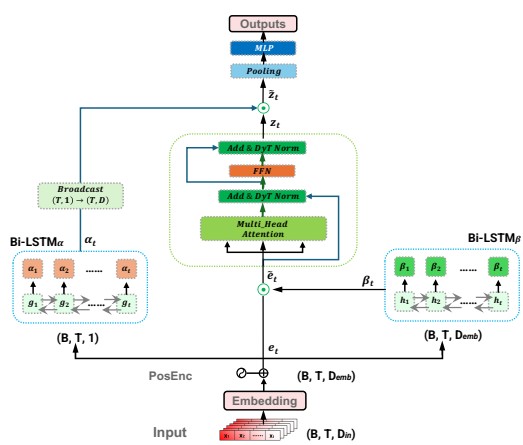

Figure 1: CoI architecture overview.

#### 3.1.2 TEMPORAL AND FEATURE-LEVEL ATTENTION

Inspired by the two-level attention framework in RETAIN Choi et al. (2016), we employ two separate bidirectional LSTM modules that operate in parallel on $\mathbf{E}_{\text{t}}$ to extract complementary temporal and feature-level importance scores. Unlike RETAIN's use of GRU units, our bidirectional LSTMs capture long-range dependencies from both past and future contexts, which is critical for detecting subtle temporal-feature interactions in clinical sequences.

**Temporal Attention Module** The first BiLSTM module processes the positionally encoded embeddings to capture temporal dynamics:

$$\mathbf{H}^{\text{temp}} = \text{BiLSTM}_\alpha(\mathbf{E}_{\text{t}}), \quad \mathbf{H}^{\text{temp}} \in \mathbb{R}^{B \times T \times 2H}, \tag{3}$$

where $H$ is the hidden dimension of each LSTM direction. The concatenated forward and backward hidden states are then projected to scalar attention scores via a linear layer followed by sigmoid activation:

$$\boldsymbol{\alpha} = \sigma(\text{Linear}_{2H \to 1}(\mathbf{H}^{\text{temp}})), \quad \boldsymbol{\alpha} \in \mathbb{R}^{B \times T \times 1}, \tag{4}$$

where $\sigma(\cdot)$ denotes the activation function. Each element $\alpha_t \in (0, 1)$ quantifies the importance of time step $t$ to the final prediction. Critically, $\boldsymbol{\alpha}$ is computed during the forward pass but applied later (see Equation 10), enabling end-to-end gradient flow.

**Feature Attention Module** In parallel, a second BiLSTM branch extracts feature-specific temporal patterns:

$$\mathbf{H}^{\text{feat}} = \text{BiLSTM}_\beta(\mathbf{E}_{\text{t}}), \quad \mathbf{H}^{\text{feat}} \in \mathbb{R}^{B \times T \times 2H}. \tag{5}$$

A separate linear projection followed by sigmoid activation produces feature-level attention vectors:

$$\boldsymbol{\beta} = \sigma(\text{Linear}_{2H \to D_{\text{emb}}}(\mathbf{H}^{\text{feat}})), \quad \boldsymbol{\beta} \in \mathbb{R}^{B \times T \times D_{\text{emb}}}, \tag{6}$$

where each $\beta_t[d] \in (0, 1)$ represents the importance of embedding dimension $d$ at time $t$. Unlike temporal attention, feature attention is applied immediately to modulate the embeddings before they enter the transformer layers.

The feature attention weights $\boldsymbol{\beta}$ are applied to $\mathbf{E}_{\text{t}}$ via element-wise multiplication:

$$\widetilde{\mathbf{E}} = \mathbf{E}_{\text{t}} \odot \boldsymbol{\beta}, \quad \widetilde{\mathbf{E}} \in \mathbb{R}^{B \times T \times D_{\text{emb}}}, \tag{7}$$

where $\odot$ denotes element-wise multiplication. This operation allows the model to selectively emphasize relevant feature dimensions before cross-feature interactions are learned in subsequent transformer layers.

### 3.1.3 CROSS-FEATURE TRANSFORMER LAYERS

The feature-weighted embeddings $\widetilde{\mathbf{E}}$ are passed through $L$ stacked Transformer layers to model temporal dependencies and inter-feature interactions. Each layer $l$ applies multi-head self-attention ($N_h$ heads) and a feed-forward network, with residual connections and Dynamic Tanh (DyT) normalization Zhu et al. (2025) after each sub-layer.

Given input $\mathbf{Z}^{(l-1)} \in \mathbb{R}^{B \times T \times D_{\text{emb}}}$ (with $\mathbf{Z}^{(0)} = \widetilde{\mathbf{E}}$), the attention output is $\mathbf{U}^{(l)}$, and attention weights from all layers are retained for interpretability (Section 3.2). The attention sub-layer is

$$\mathbf{Z}^{(l)}_{\text{attn}} = \text{DyT}\left(\mathbf{Z}^{(l-1)} + \text{Dropout}(\mathbf{U}^{(l)})\right), \tag{8}$$

where $\text{DyT}(\mathbf{x}) = \alpha \cdot \tanh(\beta \cdot \mathbf{x} + \gamma)$ with learnable $\alpha, \beta, \gamma$. The feed-forward sub-layer uses a two-layer MLP with ReLU and hidden size $4D_{\text{emb}}$:

$$\mathbf{Z}^{(l)} = \text{DyT}\left(\mathbf{Z}^{(l)}_{\text{attn}} + \text{Dropout}(\text{FFN}(\mathbf{Z}^{(l)}_{\text{attn}}))\right). \tag{9}$$

After $L$ layers, $\mathbf{Z}^{(L)} \in \mathbb{R}^{B \times T \times D_{\text{emb}}}$ encodes rich temporal–feature interactions.

### 3.1.4 TEMPORAL ATTENTION, POOLING, AND OUTPUT PROJECTION

The temporal attention weights $\boldsymbol{\alpha}$ computed in Equation 4 are applied to the transformer output via element-wise multiplication, then aggregated via global average pooling, and finally passed through MLPs to produce predictions:

$$\tilde{\mathbf{Z}} = \mathbf{Z}^{(L)} \odot \boldsymbol{\alpha}, \quad \tilde{\mathbf{Z}} \in \mathbb{R}^{B \times T \times D_{\text{emb}}}, \tag{10}$$

$$\mathbf{z} = \frac{1}{T} \sum_{t=1}^{T} \tilde{\mathbf{Z}}[:, t, :], \quad \mathbf{z} \in \mathbb{R}^{B \times D_{\text{emb}}}, \tag{11}$$

$$\hat{\mathbf{y}} = \text{MLP}(\mathbf{z}), \quad \hat{\mathbf{y}} \in \mathbb{R}^{B \times D_{\text{out}}}, \tag{12}$$

where MLP is a two-layer feed-forward network with ReLU activation.

## 3.2 INTERPRETABILITY VIA CHAIN-OF-INFLUENCE

Beyond accurate predictions, the CoI model provides rich interpretability by explicitly quantifying how features at earlier time steps influence features at later time steps. This is achieved by combining the local contribution matrix $\mathbf{C}$ (derived from dual attention mechanisms) with the cross-attention matrix $\mathbf{A}$.

### 3.2.1 LOCAL CONTRIBUTION MATRIX

The local contribution matrix $\mathbf{C} \in \mathbb{R}^{B \times T \times D_{\text{in}}}$ quantifies the importance of each input feature at every time step by integrating temporal attention weights $\boldsymbol{\alpha}$, feature-level attention vectors $\boldsymbol{\beta}$, and the embedding layer's weight matrix $\mathbf{W}_{\text{emb}}$. For each sample $n$, time step $t$, and input feature $i$:

$$C[n, t, i] = \alpha[n, t] \times \sum_{k=1}^{D_{\text{emb}}} (\beta[n, t, k] \times W_{\text{emb}}[k, i] \times X[n, t, i]), \tag{13}$$

where the summation over embedding dimensions $k$ aggregates the contribution back to the original input space. This formulation captures both the temporal importance (via $\alpha_t$) and feature-specific emphasis (via $\beta_t$), providing a fine-grained interpretation of direct feature impacts.

### 3.2.2 CROSS-ATTENTION MATRIX

To quantify how information propagates across time steps, we aggregate the attention weights from all transformer layers and heads. For each sample, layer $l$, and head $h$, the attention weights form a matrix $\mathbf{A}^{(l,h)} \in \mathbb{R}^{T \times T}$, where $A^{(l,h)}[t, t']$ represents the attention from time step $t'$ to time step $t$. We compute the averaged cross-attention matrix as:

$$A[t, t'] = \frac{1}{L \cdot N_h} \sum_{l=1}^{L} \sum_{h=1}^{N_h} A^{(l,h)}[t, t'], \tag{14}$$

where higher values of $A[t, t']$ indicate stronger temporal dependencies from time $t'$ to time $t$.

### 3.2.3 CHAIN OF INFLUENCE MEASURE

By integrating local contributions with temporal connectivity, we define a chained influence measure that quantifies how feature $i$ at time $t$ influences feature $j$ at time $t'$ (with $t < t'$):

$$\mathcal{I}(t, i; t', j) = C[t, i] \times A[t, t'] \times C[t', j]. \tag{15}$$

For efficient computation across all samples, time pairs, and feature pairs simultaneously, we implement this using vectorized Einstein summation.

## 4 EXPERIMENTS

### 4.1 DATA

We evaluate Chain-of-Influence on two complementary clinical settings that differ in both *clinical tempo* (chronic vs. acute) and *temporal structure* (regular vs. irregular sampling).

#### 4.1.1 CHRONIC KIDNEY DISEASE (CKD) DATASET

Our primary evaluation uses a proprietary longitudinal CKD cohort with scheduled nephrology follow-ups. By protocol, patients revisit every three months, yielding a regularly sampled sequence of 1,422 patients over 24 months (8 visits), with 38 features spanning demographics, comorbidities, laboratory biomarkers, and utilization. The prediction target is progression to End-Stage Renal Disease (ESRD), with 6.0% prevalence.

#### 4.1.2 MIMIC-IV DATASET

To assess generalizability in fast-changing conditions, we use MIMIC-IV v3.1 Johnson et al. (2023) and construct a cohort from first ICU stays (65,366 patients out of 94,458 stays). Raw ICU measurements are recorded at irregular times; for modeling, we aggregate events into 1-hour windows over the first 48 hours and retain irregularity signals via masks and time-gap features (time since last observation) across 15 variables (vitals, labs, demographics). The outcome is in-hospital mortality (10.8% prevalence).

Together, these datasets stress-test CoI across **chronic vs. acute** trajectories and **regular vs. irregular** temporal structure: CKD offers fixed revisit schedules for long-horizon progression, whereas MIMIC-IV captures volatile ICU dynamics with uneven sampling. Detailed cohort definitions, feature specifications, and preprocessing steps are provided in the Appendix A.

### 4.2 BASELINES

We benchmark Chain-of-Influence (CoI) against four representative models: (i) **BiLSTM**, a strong but noninterpretable recurrent baseline that processes sequences bidirectionally; (ii) **RETAIN**, the

canonical dual-attention model with visit-level ( $\alpha_t$ ) and feature-level ( $\beta_t$ ) attention; (iii) **StageNet**, which applies stageaware, channel-wise reweighting tailored to clinical progression; and (iv) **BEHRT-style Transformer**: since BEHRT was designed for tokenized medical-code inputs, we adapt its architecture backbone to our longitudinal, visit-level continuous features input settings.

### 4.3 EVALUATION METRICS

We evaluate model performance using standard classification metrics. The primary metric is the Area Under the Receiver Operating Characteristic curve (AUROC) Hanley & McNeil (1982); Fawcett (2006), which measures the model's discriminative ability across all classification thresholds. We also report the F1-Score Van Rijsbergen (1979), the harmonic mean of precision and recall that provides balanced performance assessment for imbalanced datasets. Additionally, we assess overall prediction accuracy Sokolova & Lapalme (2009), which captures the proportion of correctly classified instances across both classes. Finally, we examine precision Van Rijsbergen (1979), which measures positive predictive value and is particularly important in healthcare applications to minimize false positive predictions Powers (2011).

### 4.4 TRAINING AND IMPLEMENTATION DETAILS

Before training, we impute missing values using a MICE-based technique Raghunathan et al. (2001); Van Buuren (2007) applied separately on each dataset. Given the substantial class imbalance in both cohorts, we evaluate several mitigation strategies—including SMOTE, ADASYN, class-weighted loss, and Temporal SMOTE (TSMOTE)—and find that TSMOTE offers the best F1–AUROC trade-off (Detailed result is provided in Appendix C.2), so we adopt it for all main experiments. Hyper-parameters are selected via grid search over model-specific search spaces (Appendix B). All models are optimized with Adam and a numerically stable `BCEWithLogitsLoss`, with early stopping based on validation AUROC. We report test performance as the mean and 95% confidence interval over five random seeds. Additional training details are provided in Appendix B.6.

## 5 RESULTS & ANALYSIS

### 5.1 PREDICTIVE PERFORMANCE

Table 1 presents the comparative evaluation of CoI against four representative baselines across both clinical prediction tasks. CoI achieves state-of-the-art performance across both datasets, demonstrating the effectiveness of explicitly modeling temporal-feature influence chains. On the chronic CKD dataset, CoI attains the highest AUROC (0.960) and F1-score (0.721), outperforming the strong BEHRT-style Transformer baseline by +0.5% AUROC and +0.8% F1. While these gains may appear modest, they represent meaningful improvements over an already competitive transformer-based architecture, achieved through CoI's dual-attention mechanism that explicitly decomposes temporal and feature-level importance.

| Model | CKD Dataset | | | | | MIMIC-IV Dataset | | | | |
|---|---|---|---|---|---|---|---|---|---|---|
| | AUROC | F1 | Acc | Prec | Recall | AUROC | F1 | Acc | Prec | Recall |
| Bi-LSTM | 0.910 | 0.616 | 0.900 | 0.700 | 0.550 | 0.700 | 0.267 | 0.890 | 0.400 | 0.200 |
| RETAIN | 0.930 | 0.671 | 0.920 | 0.760 | 0.600 | 0.944 | 0.667 | 0.900 | 0.500 | **1.000** |
| StageNet | 0.940 | 0.685 | 0.930 | 0.780 | 0.610 | 0.920 | 0.768 | 0.930 | 0.700 | 0.850 |
| BEHRT-style | 0.955 | 0.715 | **0.954** | 0.810 | 0.640 | 0.948 | 0.834 | 0.945 | 0.790 | 0.884 |
| **CoI** | **0.960** | **0.721** | 0.950 | **0.820** | **0.660** | **0.950** | **0.865** | **0.950** | **0.850** | 0.880 |

Table 1: Performance comparison across datasets. Full results with confidence intervals are provided in Appendix D.1

On the acute care MIMIC-IV dataset, CoI establishes clearer superiority with AUROC of 0.950 and F1 of 0.865, improving upon BEHRT-style by +3.7% F1. A clinically critical pattern emerges when comparing CoI to RETAIN: while RETAIN achieves perfect recall (1.000), it does so at severe cost to precision (0.500), resulting in a 50% false-positive rate that would trigger unnecessary

interventions and alarm fatigue in ICU settings. CoI maintains strong recall (0.880) while achieving substantially higher precision (0.850), yielding a balanced profile essential for clinical deployment. This represents a 30% F1 improvement over RETAIN (0.865 vs. 0.667) while avoiding the trap of sensitivity-only optimization. Baseline performance reveals key architectural insights. Standard Bi-LSTM struggles dramatically with MIMIC-IV's complex temporal dynamics (F1=0.267), exposing fundamental limitations of recurrent architectures without explicit attention. StageNet provides competitive performance through stage-aware modeling but falls short of transformer-based approaches on both tasks. BEHRT-style Transformer emerges as the strongest baseline, validating self-attention's effectiveness for clinical sequences—making CoI's consistent improvements particularly significant. That CoI outperforms a strong transformer baseline suggests that its hierarchical attention—separating temporal selection, feature emphasis, and cross-feature interactions—captures richer progression patterns than vanilla self-attention, a hypothesis we examine more thoroughly via ablation study in Sec. 5.4.

## 5.2 INTERPRETABILITY ANALYSIS

### 5.2.1 ATTENTION PATTERN COMPARISON

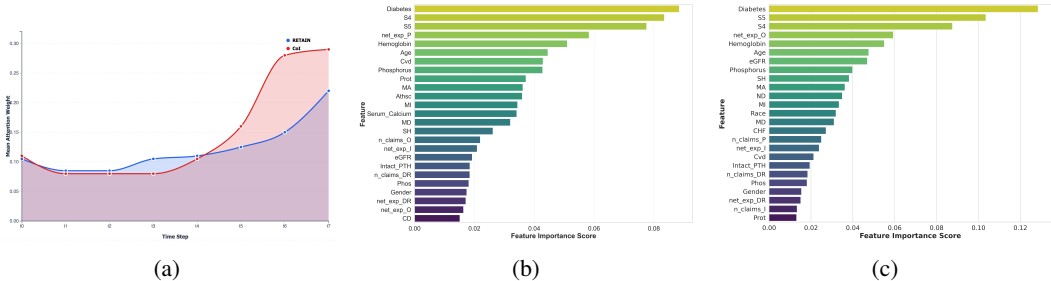

|       (a)       |       (b)       |       (c)       |

Figure 2: Temporal and feature-level importance comparison. (a) Temporal attention comparison reveals CoI's focused U-shaped pattern versus RETAIN's uniform weighting. (b-c) Feature importance profiles demonstrate consistent identification of diabetes, laboratory markers (S4, S5), and healthcare utilization as primary predictors across both models.

Figure 3 summarizes CoI's attention behavior in comparison to RETAIN on the CKD cohort. Figure 2a shows that *both* models learn a broadly U-shaped temporal importance pattern, placing higher weight on the baseline visit and on the final pre-ESRD visits. RETAIN, however, spreads mass relatively uniformly across the 8 visits, with only a mild recency bias (attention rising from roughly 0.09 at $t_1$ to 0.22 at $t_7$). CoI, in contrast, exhibits a much sharper U-shape: attention is low during the middle visits ($t_1$–$t_3$), increases steeply from $t_4$, and peaks at the last two visits ($t_6$–$t_7$, around 0.29). This pattern aligns with qualitative feedback from our collaborating nephrologists: both the baseline state and the accelerated pre-ESRD phase carry the strongest prognostic signal, whereas routine intermediate visits are less informative. CoI thus goes beyond generic recency bias and concentrates on clinically meaningful temporal windows where deterioration becomes effectively irreversible.

At the feature level (Figure 2b–2c), both models consistently identify key predictors, with diabetes emerging as the dominant risk factor, in line with clinical evidence Burckhardt et al. (2017); Li et al. (2025). Laboratory markers (S4, S5), healthcare expenditures (net_exp_P/O), and hemoglobin levels also rank among the top features. However, these static rankings obscure the temporal dynamics that CoI's influence chains reveal. For example, CoI ranks eGFR as 6th versus 14th in RETAIN, but neither ranking captures how eGFR's influence propagates through time to drive utilization patterns, or how diabetes chains through cardiovascular complications to accelerate kidney decline over specific horizons. The Chain-of-Influence paradigm addresses this gap by mapping *when* features become influential and *how* their effects cascade through clinical pathways, providing actionable insights beyond static scores.

In summary, CoI and RETAIN show consistent temporal and feature-level trends, and these patterns align with feedback from our collaborating nephrologists and clinical data providers. To more thoroughly assess the faithfulness of these attributions, we next perform a deletion-based sensitivity analysis over temporal and feature dimensions.

### 5.2.2 Sensitivity Analysis of Temporal and Feature Contributions

To assess whether CoI's temporal and feature-level attributions are aligned with its actual decision process, we conduct a deletion-based sensitivity analysis on CKD and MIMIC-IV. The key idea is that if the learned importance scores are faithful, then removing high-importance time points or features should perturb the prediction much more than removing middle- or low-importance ones. We quantify sensitivity using the absolute change in logit, $|\Delta\hat{y}|$, because the logit is directly optimized by `BCEWithLogitsLoss` and provides a non-saturated, approximately linear scale around the decision boundary, making it a more faithful indicator of how strongly an ablation perturbs the model's decision function than changes in post-sigmoid probabilities.

Table 2: Deletion-based temporal sensitivity analysis. Mean absolute change in logit $|\Delta\hat{y}|$ (mean $\pm$ 95% CI) after masking $k \in \{1, 3\}$ visits under different deletion strategies.

| **Deletion strategy** | $k$ **(visits)** | **CKD** | **MIMIC-IV** |
|---|---|---|---|
| High-attention visits | 1 | $0.42 \pm 0.03$ | $0.38 \pm 0.04$ |
| Middle-attention visits | 1 | $0.17 \pm 0.05$ | $0.15 \pm 0.02$ |
| Low-attention visits | 1 | $0.08 \pm 0.01$ | $0.09 \pm 0.02$ |
| High-attention visits | 3 | $0.89 \pm 0.06$ | $0.82 \pm 0.05$ |
| Middle-attention visits | 3 | $0.36 \pm 0.06$ | $0.32 \pm 0.03$ |
| Low-attention visits | 3 | $0.18 \pm 0.04$ | $0.16 \pm 0.04$ |

**Temporal sensitivity.** For each test trajectory, we rank visits by temporal attention $\alpha_t$ and partition them into high-, middle-, and low-attention sets. We then mask all features at the top-, middle-, or bottom-$k$ visits ($k \in \{1, 3\}$) and measure $|\Delta\hat{y}|$. As shown in Table 2, ablating high-attention visits consistently induces the largest changes: on CKD, $|\Delta\hat{y}|$ is 0.42 vs. 0.17 and 0.08 for $k = 1$, and 0.89 vs. 0.36 and 0.18 for $k = 3$. MIMIC-IV exhibits the same strict ordering with slightly smaller magnitudes. These results indicate that CoI's temporal attention focuses on time windows to which the model is genuinely most sensitive.

Table 3: Deletion-based feature sensitivity analysis. Mean absolute change in logit $|\Delta\hat{y}|$ (mean $\pm$ 95% CI) after masking $k \in \{1, 3, 5\}$ features under different deletion strategies.

| **Deletion strategy** | $k$ **(features)** | **CKD** | **MIMIC-IV** |
|---|---|---|---|
| High-attention features | 1 | $0.51 \pm 0.09$ | $0.47 \pm 0.07$ |
| Middle-attention features | 1 | $0.21 \pm 0.03$ | $0.19 \pm 0.09$ |
| Low-attention features | 1 | $0.11 \pm 0.06$ | $0.10 \pm 0.05$ |
| High-attention features | 3 | $1.02 \pm 0.03$ | $0.96 \pm 0.04$ |
| Middle-attention features | 3 | $0.43 \pm 0.04$ | $0.38 \pm 0.03$ |
| Low-attention features | 3 | $0.22 \pm 0.03$ | $0.20 \pm 0.03$ |
| High-attention features | 5 | $1.43 \pm 0.09$ | $1.34 \pm 0.08$ |
| Middle-attention features | 5 | $0.63 \pm 0.05$ | $0.57 \pm 0.04$ |
| Low-attention features | 5 | $0.31 \pm 0.03$ | $0.28 \pm 0.03$ |

**Feature sensitivity.** Analogously, we aggregate $C_{t,f}$ over time to obtain feature scores $s_f = \sum_t C_{t,f}$, rank features into high-, middle-, and low-attention groups, and mask the top-, middle-, or bottom-$k$ features across all time steps for $k \in \{1, 3, 5\}$. Table 3 shows the same monotone pattern: on CKD, removing the single highest-attention feature yields $|\Delta\hat{y}| = 0.51$ vs. 0.21 (middle) and 0.11 (low), and for $k = 5$ the gap widens to 1.43 vs. 0.63 and 0.31. MIMIC-IV again mirrors this behavior with slightly reduced magnitudes. Across both datasets and all $k$, high-attention deletions produce changes roughly 2–2.5$\times$ larger than middle-attention deletions and 4–5$\times$ larger than low-attention deletions. Taken together, these deletion experiments provide strong evidence that CoI's temporal and feature-level attributions are faithful to the model's internal decision-making, even though they do not by themselves establish causal relations in the clinical sense.

Taken together, these deletion experiments provide quantitative evidence that CoI's temporal and feature-level attributions are strongly aligned with the model's internal decision-making, even though they do not by themselves establish causal relations in the clinical sense.

## 5.3 CHAIN-OF-INFLUENCE ANALYSIS

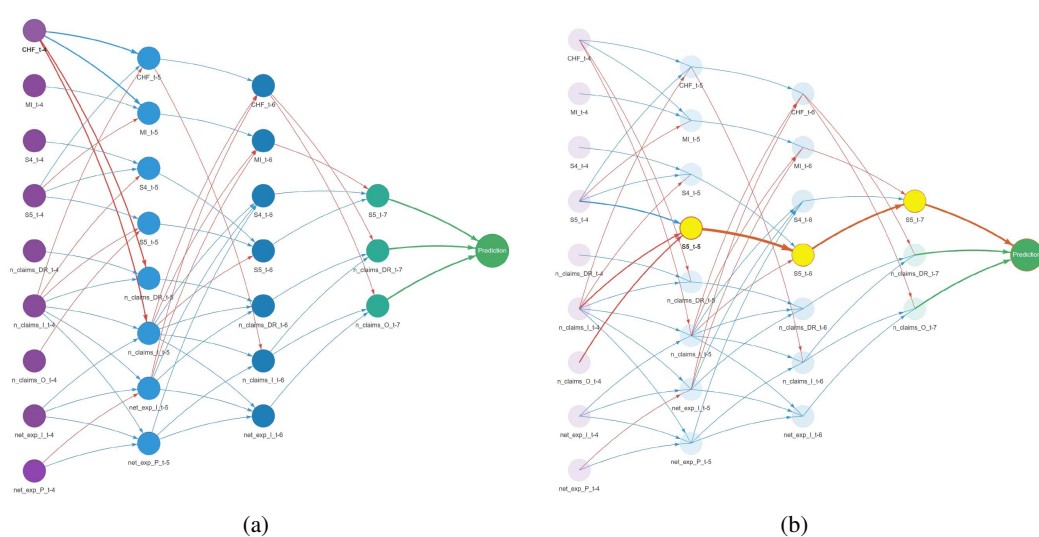

(a)                                         (b)

Figure 3: Patient-specific Chain-of-Influence visualizations. (a) Top-3 influence graph for a single patient, showing the most influential feature–time pairs and their directed paths into the prediction node. (b) Path-centric view anchored at CKD Stage 5 at $t_5$, highlighting the high-influence chain through subsequent visits that drives the final ESRD risk. Additional visualization examples are provided in Appendix E.

Figure 3a visualizes the patient–specific Chain-of-Influence graph induced by Eq. equation 15 for a single CKD patient after the observation window. Each node corresponds to a feature–time pair $(i, t)$ and nodes are arranged from left (earlier visits) to right (later visits and the prediction node). Each edge encodes the magnitude of $\mathcal{I}(t, i; t', j)$. Because the full graph is dense, we introduce a *top-k* sparsification scheme for visualization: at the final visit $t_7$ we retain only the $k$ most influential incoming feature–time pairs; for each of these nodes we again retain only their top-$k$ predecessors at $t_6$, and so on recursively. Panel 3a shows the resulting sparse, time–unfolded influence network for $k = 3$, $T_{start} = t_4$, revealing a small set of clinically influence pathways that contribute to the patient's risk prediction.

To support path–centric reasoning, we also trace the forward influence of a *single* clinically meaningful starting point. In Figure 3b, we anchor the visualization on CKD Stage 5 at visit $t_5$ and highlight only nodes and edges on high–influence paths emanating from this feature. Background nodes and edges are faded, while the active chain is emphasized in yellow (nodes) and orange (edges). This view reveals how advanced CKD at $t_5$ propagates through subsequent utilization and severity markers at $t_6$ and $t_7$, and ultimately into the prediction node. Clinicians can thus read off a concrete, patient–level narrative; for example, the Stage 5 $t_5 \to t_6 \to t_7$ progression exemplifies a self–reinforcing temporal chain in which each occurrence amplifies later disease burden and progression risk—consistent with the clinical reality that Stage 5 CKD is effectively irreversible and a dominant driver of ESRD outcome.

Together, these graph–centric (top-$k$) and path–centric views demonstrate how CoI turns the abstract influence tensor into structured, human–readable stories of full mapping leading to the predictions. Additional visualization examples are provided in Appendix E.

## 5.4 ABLATION STUDY

To validate our architectural design choices, we conduct an ablation study over seven CoI variants (Figure 4). Removing any component substantially degrades performance on both datasets. Transformer layers are the most critical: their removal yields the largest F1 drop on CKD (-0.101) and the second-largest on MIMIC-IV (-0.100), supporting our choice to model cross-feature interactions via self-attention. The ablations also reveal a clear dataset-specific pattern: feature attention is more important for chronic CKD progression (CKD: ΔAUROC -0.065), whereas temporal attention is more important for acute ICU mortality (MIMIC-IV: ΔAUROC -0.090). This matches clinical intuition—CKD risk is driven by *which* biomarkers deteriorate, while ICU outcomes depend more on *when* instability occurs.

Component synergy proves essential—the "Temporal only" and "Feature only" variants underperform the full model by 5-10% in F1-score, while the "Transformer Only" baseline shows the worst degradation (CKD F1: -0.116, MIMIC-IV F1: -0.135). This demonstrates that self-attention alone cannot replicate the structured temporal-feature decomposition provided by our dual attention pathways. The full CoI model's superior performance emerges from the multiplicative integration of attention mechanisms (Equations 7, 10), enabling feature attention to guide what infor-

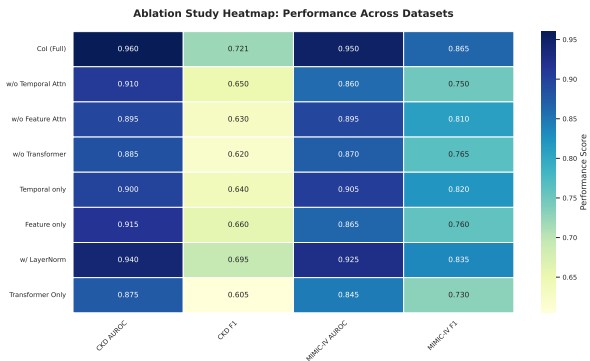

Figure 4: CoI architecture overview. Detailed results are provided in Appendix D.2

mation enters the transformer while temporal attention weights the final representation. Replacing DyT with standard LayerNorm reduces performance uniformly (CKD: -2.6% F1, MIMIC-IV: -3.0% F1), validating DyT's adaptive normalization for clinical data's non-stationary distributions.

## 6 CONCLUSION

We introduced Chain-of-Influence (CoI), a deep learning framework that advances clinical predictive modeling by explicitly capturing and visualizing temporal-feature interdependencies. Through a hierarchical architecture integrating temporal attention, feature-level attention, and cross-feature transformer layers, CoI achieves state-of-the-art performance across heterogeneous clinical settings—attaining 0.960 AUROC on chronic kidney disease progression and 0.950 AUROC on ICU mortality prediction. Our ablation studies demonstrate that these gains emerge from the synergistic integration of attention mechanisms, with feature attention proving more critical for chronic disease and temporal attention dominating in acute care. Deletion-based sensitivity analyses further validate that CoI's learned attributions faithfully reflect its internal decision process.

Beyond predictive accuracy, CoI transforms black-box predictions into structured, patient-specific influence chains that mirror clinical reasoning. By mapping not only *what* features matter but *when* they become influential and *how* their effects cascade through clinical pathways, CoI provides actionable insights that transcend static importance scores—offering clinicians a transparent and generalizable framework for understanding the complex interplay of clinical variables over time.

**Limitations** We acknowledge key limitations: the learned influence chains represent powerful statistical associations, not formal causal pathways, and our aggregation of attention weights may obscure finer-grained component roles. Furthermore, the framework's $O\left(T^2 \cdot F^2\right)$ computational complexity presents a scalability challenge for datasets with extremely long time series or numerous features. Future work could address these by integrating causal discovery methods and exploring sparse approximation techniques.

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

## A DATASET CHARACTERISTICS

### A.1 CHRONIC KIDNEY DISEASE DATASET STATISTICS

COHORT OVERVIEW AND COMPARATIVE ANALYSIS

The Chronic Kidney Disease (CKD) dataset represents a comprehensive longitudinal study of 1,422 patients with progressive kidney disease, followed over 24 months. Table 4 presents a comparative analysis between patients who progressed to ESRD versus those who did not, providing critical insights into risk factors and disease progression patterns.

Table 4: CKD Dataset: Comprehensive Patient Characteristics by ESRD Progression Status

| Characteristic | ESRD Progression (n=86, 6.0%) | No ESRD Progression (n=1,336, 94.0%) | P-value |
|---|---|---|---|
| **Demographics** | | | |
| Age (years) | $69.13 \pm 12.37$ | $72.04 \pm 11.25$ | $< 0.001$ |
| Female | 40 (46.5%) | 721 (54.0%) | 0.215 |
| Race | | | $< 0.001$ |
| White | 70 (81.4%) | $1,242$ (93.0%) | |
| African American | 12 (14.0%) | 60 (4.5%) | |
| Other | 4 (4.6%) | 34 (2.5%) | |
| BMI (kg/m²) | $28.40 \pm 5.32$ | $26.40 \pm 6.20$ | $< 0.001$ |
| **Comorbidities** | | | |
| Diabetes | 63 (73.3%) | 788 (59.0%) | 0.009 |
| Hypertension | 85 (99.0%) | $1,323$ (99.0%) | 0.863 |
| Cardiovascular Disease | 10 (17.2%) | 177 (18.2%) | 0.990 |
| Anemia | 55 (64.0%) | 828 (62.0%) | 0.714 |
| Metabolic Acidosis | 22 (25.6%) | 240 (18.0%) | 0.077 |
| Proteinuria | 11 (12.8%) | 227 (17.0%) | 0.312 |
| Secondary Hyperparathyroidism | 28 (32.6%) | 240 (18.0%) | $< 0.001$ |
| Phosphatemia | 4 (4.7%) | 40 (3.0%) | 0.390 |
| Atherosclerosis | 6 (9.4%) | 149 (14.6%) | 0.320 |
| Heart Failure | 6 (7.0%) | 120 (9.0%) | 0.526 |
| Stroke | 1 (1.2%) | 40 (3.0%) | 0.506 |
| Conduction & Dysrhythmias | 4 (4.7%) | 214 (16.0%) | 0.005 |
| Myocardial Infarction | 31 (51.7%) | 316 (32.4%) | 0.003 |
| Fluid & Electrolyte Disorders | 9 (17.6%) | 122 (13.3%) | 0.503 |
| Metabolic Disorders | 5 (9.4%) | 60 (6.5%) | 0.581 |
| Nutritional Disorders | 6 (10.2%) | 106 (11.6%) | 0.900 |
| CKD Stage 4 | 47 (54.7%) | 298 (22.3%) | $< 0.001$ |
| CKD Stage 5 | 42 (48.8%) | 277 (20.7%) | $< 0.001$ |

FEATURE CATEGORIES AND DESCRIPTIONS

The 38 clinical features in the CKD dataset are systematically organized across four major categories, each capturing distinct aspects of patient health and disease progression.

Table 5: CKD Dataset: Complete Feature Specifications

| Category | Feature | Description |
|---|---|---|
| **Demographics** | Age | Patient age at baseline (years) |
| | Gender | Binary indicator (Male=0, Female=1) |
| | Race | Categorical encoding (White, Black, Hispanic, Other) |
| | BMI | Body Mass Index (kg/m²) |
| **Comorbidities** | Diabetes | Type 1 or Type 2 diabetes mellitus diagnosis |
| | Htn | Hypertension (systolic $\geq$140 or diastolic $\geq$90 mmHg) |

Table 5 – continued from previous page

| Category | Feature | Description |
|---|---|---|
| | Cvd | Cardiovascular disease (coronary, cerebrovascular, peripheral) |
| | Anemia | Hemoglobin <13 g/dL (men) or <12 g/dL (women) |
| | MA | Mineral and bone disorders |
| | Prot | Proteinuria (>300 mg/day or >300 mg/g creatinine) |
| | SH | Secondary hyperparathyroidism |
| | Phos | Phosphorus disorders (hyperphosphatemia) |
| | Athsc | Atherosclerosis |
| | CHF | Congestive heart failure |
| | Stroke | Cerebrovascular accident history |
| | CD | Coronary disease |
| | MI | Myocardial infarction history |
| | FE | Iron deficiency |
| | MD | Metabolic disorders |
| | ND | Nutritional deficiency |
| | S4 | CKD Stage 4 (eGFR 15-29 mL/min/1.73m²) |
| | S5 | CKD Stage 5 (eGFR <15 mL/min/1.73m²) |
| Lab Biomarkers | Serum_Calcium | Serum calcium levels (mg/dL) |
| | eGFR | Estimated glomerular filtration rate (mL/min/1.73m²) |
| | Phosphorus | Serum phosphorus levels (mg/dL) |
| | Intact_PTH | Intact parathyroid hormone (pg/mL) |
| | Hemoglobin | Blood hemoglobin concentration (g/dL) |
| | UACR | Urine albumin-to-creatinine ratio (mg/g) |
| | Bicarbonate | Serum bicarbonate levels (mEq/L) |
| Healthcare Utilization | n_claims_DR | Number of durable medical equipment/drug claims |
| | n_claims_I | Number of inpatient claims |
| | n_claims_O | Number of outpatient claims |
| | n_claims_P | Number of physician service claims |
| | net_exp_DR | Net expenditure for DME/drugs ($) |
| | net_exp_I | Net expenditure for inpatient services ($) |
| | net_exp_O | Net expenditure for outpatient services ($) |
| | net_exp_P | Net expenditure for physician services ($) |

STATISTICAL ANALYSIS AND CLINICAL SIGNIFICANCE

**Cohort Characteristics:** The complete cohort comprises 1,422 CKD patients with 24-month follow-up, yielding 8 temporal observations per patient (3-month intervals). ESRD progression occurred in 86 patients (6.0%), representing a clinically realistic progression rate for this advanced CKD population.

**Significant Risk Factors (p $< 0.05$):** Statistical analysis revealed several key differentiators between ESRD progressors and non-progressors:

*Demographic Factors:*

- Younger age (69.1 vs 72.0 years) - counterintuitive finding suggesting more aggressive disease in younger patients
- Higher BMI (28.4 vs 26.4 kg/m²) - indicating metabolic burden
- Higher proportion of African Americans (14.0% vs 4.5%) - known ethnic disparity in CKD progression

*Comorbidity Profile:*

- Higher diabetes prevalence (73.3% vs 59.0%) - primary driver of CKD progression

Table 6: CKD Dataset: Laboratory Values and Healthcare Utilization by ESRD Progression Status

| Feature | ESRD Progression (n=86) | No ESRD Progression (n=1,336) | P-value |
|---|---|---|---|
| **Laboratory Biomarkers** | | | |
| eGFR (mL/min/1.73m²) | $17.21 \pm 5.46$ | $22.78 \pm 5.66$ | $< 0.001$ |
| Hemoglobin (g/dL) | $12.15 \pm 2.19$ | $14.25 \pm 1.80$ | $< 0.001$ |
| Bicarbonate (mEq/L) | $22.90 \pm 6.36$ | $25.30 \pm 4.22$ | 0.001 |
| Serum Calcium (mg/dL) | $9.39 \pm 3.62$ | $10.21 \pm 2.86$ | 0.042 |
| Phosphorus (mg/dL) | $3.61 \pm 0.87$ | $3.52 \pm 0.72$ | 0.350 |
| Intact PTH (pg/mL) | $78.66 \pm 40.23$ | $62.72 \pm 37.32$ | 0.001 |
| **Healthcare Utilization Claims** | | | |
| Pharmacy Claims (count) | $120 \pm 94$ | $109 \pm 86$ | 0.293 |
| Inpatient Claims (count) | $3.85 \pm 3.41$ | $3.74 \pm 3.62$ | 0.773 |
| Outpatient Claims (count) | $27.78 \pm 24.75$ | $22.07 \pm 19.13$ | 0.039 |
| Professional Claims (count) | $105.37 \pm 77.56$ | $87.43 \pm 68.02$ | 0.039 |
| **Healthcare Expenditures ($)** | | | |
| Pharmacy Expenditure | $12,053 \pm 11,596$ | $10,440 \pm 20,662$ | 0.242 |
| Inpatient Expenditure | $33,909 \pm 53,540$ | $29,440 \pm 32,541$ | 0.446 |
| Outpatient Expenditure | $9,354 \pm 17,522$ | $8,554 \pm 17,492$ | 0.682 |
| Professional Expenditure | $15,512 \pm 18,657$ | $11,640 \pm 12,748$ | 0.061 |

- Increased secondary hyperparathyroidism (32.6% vs 18.0%) - marker of mineral bone disorder

- More myocardial infarction (51.7% vs 32.4%) - cardiovascular complications

- Paradoxically lower conduction disorders (4.7% vs 16.0%) - may reflect survival bias

- Higher CKD stage 4 (54.7% vs 22.3%) and stage 5 (48.8% vs 20.7%) prevalence

*Laboratory Markers:*

- Significantly lower eGFR (17.2 vs 22.8 mL/min/1.73m²) - expected primary marker

- Lower hemoglobin (12.2 vs 14.3 g/dL) - indicating CKD-related anemia

- Lower bicarbonate (22.9 vs 25.3 mEq/L) - metabolic acidosis marker

- Higher intact PTH (78.7 vs 62.7 pg/mL) - secondary hyperparathyroidism

- Lower serum calcium (9.4 vs 10.2 mg/dL) - mineral bone disorder

*Healthcare Utilization:*

- Higher outpatient claims (27.8 vs 22.1) - increased monitoring needs

- Higher professional claims (105.4 vs 87.4) - specialist consultations

## A.2 MIMIC-IV DATASET CHARACTERISTICS

### COHORT CONSTRUCTION AND SELECTION CRITERIA

The MIMIC-IV cohort was constructed through a systematic pipeline designed to ensure data quality and clinical relevance for mortality prediction tasks.

**Initial Database Statistics:**

- Total ICU stays in MIMIC-IV: 94,458

- Unique patients: 65,366

- Multiple ICU stays per patient: 29,092 additional stays

**Cohort Construction:** Our preprocessing pipeline selects the first ICU stay per patient to ensure independence of observations, resulting in exactly 65,366 patients. This approach eliminates potential confounding from repeated admissions while maintaining a substantial sample size for robust model training and evaluation.

**Final Cohort:** 65,366 patients meeting all inclusion criteria.

FEATURE STATISTICS

Table 7 presents detailed statistics for all MIMIC-IV features, including percentile distributions to capture the full range of physiological values encountered in critical care.

Table 7: MIMIC-IV Dataset: Comprehensive Feature Statistics (n=65,366 patients)

| Feature | Mean ± SD | Median [IQR] | Missing (%) |
|---|---|---|---|
| **Demographics** | | | |
| Age (years) | $64.5 \pm 17.1$ | $66.0[54.0, 78.0]$ | 0.0 |
| Gender (Female %) | 43.8% | - | 0.0 |
| **Vital Signs** | | | |
| Heart Rate (bpm) | $84.1 \pm 18.9^*$ | $82.0[71.0, 95.0]$ | 27.5 |
| Systolic BP (mmHg) | $119.9 \pm 25.8^*$ | $117.0[104.0, 133.0]$ | 51.6 |
| Diastolic BP (mmHg) | $67.4 \pm 15.8^*$ | $64.0[55.0, 75.0]$ | 51.6 |
| Mean BP (mmHg) | $87.6 \pm 16.9^*$ | $78.0[69.0, 89.0]$ | 51.6 |
| Respiratory Rate (rpm) | $22.7 \pm 6.8^*$ | $19.0[15.0, 22.0]$ | 28.4 |
| Temperature (°C) | $37.0 \pm 0.8$ | $36.8[36.6, 37.2]$ | 81.3 |
| SpO2 (%) | $96.8 \pm 3.2^*$ | $97.0[95.0, 99.0]$ | 28.9 |
| **Laboratory Tests** | | | |
| Creatinine (mg/dL) | $1.4 \pm 1.5$ | $1.0[0.7, 1.5]$ | 93.1 |
| Glucose (mg/dL) | $141.1 \pm 69.7$ | $125.0[103.0, 157.0]$ | 93.5 |
| Sodium (mEq/L) | $138.2 \pm 5.6$ | $138.0[135.0, 141.0]$ | 92.9 |
| Potassium (mEq/L) | $4.2 \pm 0.7$ | $4.1[3.8, 4.5]$ | 92.8 |
| Hematocrit (%) | $31.1 \pm 6.0$ | $30.5[26.7, 35.1]$ | 92.3 |
| WBC (K/μL) | $12.3 \pm 10.7$ | $10.7[7.8, 14.7]$ | 93.3 |

*Note: * indicates features with outliers present in raw data that require preprocessing. Standard deviations shown are after outlier detection but before winsorization.*

**Outlier Detection and Treatment:** Physiologically implausible values are identified using clinical knowledge-based thresholds and statistical methods:

- Heart Rate: $<20$ or $>220$ bpm
- Blood Pressure: Systolic $<50$ or $>250$ mmHg, Diastolic $<20$ or $>150$ mmHg
- Temperature: $<32$°C or $>42$°C
- SpO2: $<70\%$ or $>100\%$
- Laboratory values: Beyond 99.5th percentile of physiologically reasonable ranges

Outliers are winsorized to the 1st and 99th percentiles to preserve distributional properties while reducing the impact of measurement errors.

**Normalization:** All continuous features are standardized using z-score normalization: $z = \frac{x-\mu}{\sigma}$, where $\mu$ and $\sigma$ are computed on the training set only to prevent data leakage.

COHORT CHARACTERISTICS AND OUTCOMES

Table 8: MIMIC-IV Cohort: Clinical Characteristics and Outcomes

| Characteristic | Value | Percentage |
|---|---|---|
| Total Patients | 65,366 | - |
| Age Distribution | | |
|    Mean Age | $64.5 \pm 17.1$ years | - |
|    Age Range | 18-103 years | - |
| Gender | | |
|    Male | 36,720 | 56.2% |
|    Female | 28,646 | 43.8% |
| Temporal Structure | | |
|    Observation Window | 48 hours | Hourly resolution |
|    Total Observations | 3,137,568 | $(65,366 \times 48)$ |
|    Features per Patient | 15 | 7 vitals + 6 labs + 2 demographics |
| **Clinical Outcomes** | | |
| Hospital Mortality | 7,086 | 10.8% |
| Survivors | 58,280 | 89.2% |
| **Data Quality Characteristics** | | |
| Vital Signs Missing Data | 27.5-81.3% | Variable by feature type |
| Laboratory Tests Missing Data | 92.3-93.5% | Expected for ICU setting |
| Demographics Missing Data | 0.0% | Complete for all patients |

# B    HYPERPARAMETER CONFIGURATION DETAILS

This appendix provides comprehensive hyperparameter details for five key models evaluated on the Chronic Kidney Disease (CKD) progression prediction task: BiLSTM, RETAIN, StageNet, BEHRT-style Transformer, and Chain-of-Influence (CoI). For each model, we present both the search space of hyperparameter candidates and the optimal configuration identified through hyperparameter optimization.

## B.1    BiLSTM (BIDIRECTIONAL LONG SHORT-TERM MEMORY)

HYPERPARAMETER SEARCH SPACE

| Hyperparameter | Candidates |
|---|---|
| Batch Size | {32, 64, 128} |
| Dropout Rate | {0.1, 0.2, 0.3} |
| Hidden Dimension | {32, 64, 128} |
| Learning Rate | {0.0001, 0.001, 0.01} |
| Number of Layers | {1, 2, 3} |

Table 9: BiLSTM hyperparameter search space (Total: $3 \times 3 \times 3 \times 3 \times 3 = 243$ combinations).

OPTIMAL CONFIGURATION

| Hyperparameter | Best Value |
|---|---|
| Batch Size | 128 |
| Dropout Rate | 0.2 |
| Hidden Dimension | 32 |
| Learning Rate | 0.001 |
| Number of Layers | 2 |

Table 10: BiLSTM optimal hyperparameters.

## B.2 RETAIN (REVERSE TIME ATTENTION)

HYPERPARAMETER SEARCH SPACE

| Hyperparameter | Candidates |
|---|---|
| Batch Size | {32, 64, 128} |
| Dropout Rate | {0.0, 0.1, 0.2} |
| Embedding Dimension | {16, 32, 64} |
| Hidden Dimension | {32, 64, 128} |
| Learning Rate | {0.0001, 0.001, 0.01} |
| Number of Attention Heads | {2, 4, 8, 16} |
| Number of Layers | {2, 3, 4} |

Table 11: RETAIN hyperparameter search space (Total: $3^6 \times 4 = 2916$ combinations).

OPTIMAL CONFIGURATION

| Hyperparameter | Best Value |
|---|---|
| Batch Size | 64 |
| Dropout Rate | 0.0 |
| Embedding Dimension | 16 |
| Hidden Dimension | 32 |
| Learning Rate | 0.001 |
| Number of Attention Heads | 8 |
| Number of Layers | 3 |

Table 12: RETAIN optimal hyperparameters.

## B.3 STAGENET (STAGE-AWARE NEURAL NETWORK)

HYPERPARAMETER SEARCH SPACE

| Hyperparameter | Candidates |
|---|---|
| Batch Size | {32, 64, 128} |
| Dropout Rate | {0.1, 0.2, 0.3} |
| Hidden Dimension | {128, 256, 512} |
| Learning Rate | {0.0001, 0.001, 0.01} |
| Number of Stages | {3, 4, 5} |
| Kernel Size | {3, 5, 7} |

Table 13: StageNet hyperparameter search space (Total: $3^6 = 729$ combinations).

OPTIMAL CONFIGURATION

| Hyperparameter | Best Value |
|---|---|
| Batch Size | 128 |
| Dropout Rate | 0.2 |
| Hidden Dimension | 512 |
| Learning Rate | 0.0001 |
| Number of Stages | 5 |
| Kernel Size | 5 |

Table 14: StageNet optimal hyperparameters.

## B.4 BEHRT-STYLE TRANSFORMER

HYPERPARAMETER SEARCH SPACE

| Hyperparameter | Candidates |
|---|---|
| Batch Size | $\{32, 64, 128\}$ |
| Dropout Rate | $\{0.1, 0.2, 0.3\}$ |
| Embedding Dimension | $\{16, 32, 64\}$ |
| Hidden Dimension | $\{64, 128, 256\}$ |
| Learning Rate | $\{0.0001, 0.001\}$ |
| Number of Attention Heads | $\{2, 4, 8\}$ |
| Number of Layers | $\{2, 4, 6\}$ |

Table 15: BEHRT-style Transformer hyperparameter search space (Total: $3 \times 3 \times 3 \times 3 \times 2 \times 3 \times 3 = 1458$ combinations).

OPTIMAL CONFIGURATION

| Hyperparameter | Best Value |
|---|---|
| Batch Size | 128 |
| Dropout Rate | 0.2 |
| Embedding Dimension | 32 |
| Hidden Dimension | 128 |
| Learning Rate | 0.0001 |
| Number of Attention Heads | 4 |
| Number of Layers | 4 |

Table 16: BEHRT-style Transformer optimal hyperparameters.

## B.5 CoI (CHAIN-OF-INFLUENCE WITH DYNAMIC TANH NORMALIZATION)

HYPERPARAMETER SEARCH SPACE

| Hyperparameter | Candidates |
|---|---|
| Batch Size | $\{32,64,128\}$ |
| Dropout Rate | $\{0.0, 0.2, 0.4\}$ |
| Embedding Dimension | $\{16, 32, 64\}$ |
| Hidden Dimension | $\{32, 64, 128\}$ |
| Learning Rate | $\{0.001, 0.0001\}$ |
| Number of Attention Heads | $\{2, 4, 8\}$ |
| Number of Layers | $\{4, 6, 8\}$ |
| Alpha Initialization (DyT) | $\{0.2, 0.4, 0.6, 0.8\}$ |

Table 17: CoI hyperparameter search space (Total: $3\times3\times3\times3\times2\times3\times3\times4 = 5832$ combinations).

OPTIMAL CONFIGURATION

| Hyperparameter | Best Value |
|---|---|
| Batch Size | 128 |
| Dropout Rate | 0.2 |
| Embedding Dimension | 16 |
| Hidden Dimension | 128 |
| Learning Rate | 0.0001 |
| Number of Attention Heads | 2 |
| Number of Layers | 8 |
| Alpha Initialization (DyT) | 0.8 |

Table 18: CoI optimal hyperparameters.

## B.6 TRAINING CONFIGURATION

All models were trained under a common protocol:

| Parameter | Value |
|---|---|
| Maximum Epochs | 500 |
| Early Stopping Patience | 10 (on validation F1 ) |
| Optimizer | Adam |
| Loss Function | BCEWithLogitsLoss |
| Validation Split | 20% of training data |
| Test Split | 20% (held-out) |
| Number of Random Seeds | 5 |

Table 19: Common training configuration for all models.

## C    TRAINING DETAILS

### C.1    COMPUTATIONAL COST AND TRAINING TIME

All experiments were conducted on a workstation equipped with $2\times$NVIDIA RTX 4090 GPUs (24GB each). Table 20 reports the average wall-clock training time per run (including early stopping) for each model on the CKD and MIMIC-IV tasks. Times are averaged over the five random seeds used in our main experiments; individual runs exhibit small variability ($\pm 5$–10 minutes) depending on early-stopping epoch.

| Model | CKD (hours/run) | MIMIC-IV (hours/run) |
|---|---|---|
| BiLSTM | 0.10 | 0.25 |
| RETAIN | 0.15 | 0.40 |
| StageNet | 0.30 | 0.90 |
| BEHRT-style Transformer | 0.35 | 1.00 |
| **CoI** | **0.40** | **1.30** |

Table 20: Average wall-clock training time per run on $2\times$RTX 4090 GPUs. Each value corresponds to one full training run with early stopping for a single random seed.

Hyperparameter tuning was performed using the grid search spaces listed in the previous subsections, combined with early stopping. In practice, we evaluated up to 40–60 candidate configurations per model on CKD (fewer for heavier architectures such as CoI and the BEHRT-style Transformer), and then retrained the selected best configuration for five seeds. Across all five models and the CoI ablations, the total compute budget amounted to approximately 220 GPU-hours, corresponding to roughly 110 hours of wall-clock time on $2\times$RTX 4090 GPUs. This budget includes both hyperparameter search and final multi-seed training runs for CKD and MIMIC-IV.

## C.2 COMPARISON OF CLASS-IMBALANCE STRATEGIES

Both datasets exhibit substantial class imbalance—CKD progression (6.0% positive class) and MIMIC-IV mortality (10.8% positive class)—which can significantly affect both optimization and calibration of predictive models. To justify our imbalance-handling approach, we compare several commonly used strategies applied to CoI on both datasets:

- **None**: no explicit rebalancing beyond the original class distribution.
- **SMOTE**: standard Synthetic Minority Oversampling Technique applied at the patient-level.
- **ADASYN**: adaptive synthetic oversampling that focuses on harder minority examples.
- **ENN**: Edited Nearest Neighbors undersampling of majority examples near the decision boundary.
- **SMOTE+ENN**: sequential SMOTE oversampling followed by ENN undersampling.
- **Weighted Loss**: class-weighted BCEWithLogitsLoss with inverse-frequency weights.
- **TSMOTE**: temporal SMOTE variant that interpolates entire minority trajectories while preserving time ordering.

Table 21 reports AUROC and F1 for each strategy on CKD. All results are averaged over five random seeds using the same architecture and optimization settings as in the main CoI experiments.

| Imbalance Strategy | CKD Dataset | | MIMIC-IV Dataset | |
|---|---|---|---|---|
| | AUROC ($\pm$CI) | F1 ($\pm$CI) | AUROC ($\pm$CI) | F1 ($\pm$CI) |
| None | $0.930 \pm 0.020$ | $0.580 \pm 0.030$ | $0.920 \pm 0.018$ | $0.780 \pm 0.025$ |
| SMOTE | $0.945 \pm 0.018$ | $0.640 \pm 0.026$ | $0.935 \pm 0.016$ | $0.820 \pm 0.022$ |
| ADASYN | $0.942 \pm 0.019$ | $0.630 \pm 0.028$ | $0.932 \pm 0.017$ | $0.815 \pm 0.023$ |
| ENN | $0.938 \pm 0.022$ | $0.600 \pm 0.027$ | $0.928 \pm 0.019$ | $0.800 \pm 0.024$ |
| SMOTE+ENN | $0.952 \pm 0.016$ | $0.690 \pm 0.022$ | $0.942 \pm 0.014$ | $0.840 \pm 0.020$ |
| Weighted Loss | $0.955 \pm 0.015$ | $0.700 \pm 0.020$ | $0.945 \pm 0.013$ | $0.850 \pm 0.018$ |
| **TSMOTE (ours)** | $\mathbf{0.960 \pm 0.010}$ | $\mathbf{0.721 \pm 0.013}$ | $\mathbf{0.950 \pm 0.009}$ | $\mathbf{0.865 \pm 0.012}$ |

Table 21: Comparison of class-imbalance handling strategies for CoI on both datasets. Values denote mean $\pm$ 95% confidence intervals over 5 runs. Naive training without rebalancing (*None*) performs substantially worse with higher variance, while TSMOTE yields the best F1-AUROC trade-off on both datasets and is therefore used in all main experiments.

# D Experimental Result Details

## D.1 Performance comparison details across datasets

| **CKD Dataset** | | | | | |
|---|---|---|---|---|---|
| Model | AUROC ($\pm$CI) | F1 ($\pm$CI) | Acc ($\pm$CI) | Prec ($\pm$CI) | Recall ($\pm$CI) |
| Bi-LSTM | $0.910 \pm 0.012$ | $0.616 \pm 0.018$ | $0.900 \pm 0.011$ | $0.700 \pm 0.016$ | $0.550 \pm 0.018$ |
| RETAIN | $0.930 \pm 0.011$ | $0.671 \pm 0.017$ | $0.920 \pm 0.010$ | $0.760 \pm 0.015$ | $0.600 \pm 0.017$ |
| StageNet | $0.940 \pm 0.010$ | $0.685 \pm 0.016$ | $0.930 \pm 0.010$ | $0.780 \pm 0.014$ | $0.610 \pm 0.016$ |
| BEHRT-style | $0.955 \pm 0.010$ | $0.715 \pm 0.015$ | $\mathbf{0.954 \pm 0.009}$ | $0.810 \pm 0.013$ | $0.640 \pm 0.017$ |
| **CoI** | $\mathbf{0.960 \pm 0.009}$ | $\mathbf{0.721 \pm 0.013}$ | $0.950 \pm 0.008$ | $\mathbf{0.820 \pm 0.011}$ | $\mathbf{0.660 \pm 0.016}$ |

| **MIMIC-IV Dataset** | | | | | |
|---|---|---|---|---|---|
| Model | AUROC ($\pm$CI) | F1 ($\pm$CI) | Acc ($\pm$CI) | Prec ($\pm$CI) | Recall ($\pm$CI) |
| Bi-LSTM | $0.700 \pm 0.018$ | $0.267 \pm 0.018$ | $0.890 \pm 0.012$ | $0.400 \pm 0.017$ | $0.200 \pm 0.018$ |
| RETAIN | $0.944 \pm 0.013$ | $0.667 \pm 0.018$ | $0.900 \pm 0.011$ | $0.500 \pm 0.016$ | $\mathbf{1.000 \pm 0.000}$ |
| StageNet | $0.920 \pm 0.012$ | $0.768 \pm 0.017$ | $0.930 \pm 0.010$ | $0.700 \pm 0.015$ | $0.850 \pm 0.017$ |
| BEHRT-style | $0.948 \pm 0.011$ | $0.834 \pm 0.016$ | $0.945 \pm 0.009$ | $0.790 \pm 0.014$ | $0.884 \pm 0.017$ |
| **CoI** | $\mathbf{0.950 \pm 0.010}$ | $\mathbf{0.865 \pm 0.015}$ | $\mathbf{0.950 \pm 0.009}$ | $\mathbf{0.850 \pm 0.013}$ | $0.880 \pm 0.016$ |

Table 22: Performance comparison on CKD and MIMIC-IV datasets. Values denote mean $\pm$ 95% confidence intervals over 5 runs (per-metric standard deviations $<$ 0.015). Best results per column are highlighted in bold.

## D.2 Ablation Study Result Details

| **Model Variant** | **CKD Dataset** | | | | **MIMIC-IV Dataset** | | | |
|---|---|---|---|---|---|---|---|---|
| | AUROC ($\pm$CI) | $\Delta$ | F1 ($\pm$CI) | $\Delta$ | AUROC ($\pm$CI) | $\Delta$ | F1 ($\pm$CI) | $\Delta$ |
| **CoI (Full)** | $\mathbf{0.960 \pm 0.010}$ | - | $\mathbf{0.721 \pm 0.013}$ | - | $\mathbf{0.950 \pm 0.009}$ | - | $\mathbf{0.865 \pm 0.012}$ | - |
| w/o Temporal Attn | $0.910 \pm 0.011$ | -0.050 | $0.650 \pm 0.014$ | -0.071 | $0.860 \pm 0.012$ | -0.090 | $0.750 \pm 0.015$ | -0.115 |
| w/o Feature Attn | $0.895 \pm 0.010$ | -0.065 | $0.630 \pm 0.015$ | -0.091 | $0.895 \pm 0.011$ | -0.055 | $0.810 \pm 0.013$ | -0.055 |
| w/o Transformer | $0.885 \pm 0.012$ | -0.075 | $0.620 \pm 0.015$ | -0.101 | $0.870 \pm 0.013$ | -0.080 | $0.765 \pm 0.016$ | -0.100 |
| Temporal only | $0.900 \pm 0.011$ | -0.060 | $0.640 \pm 0.013$ | -0.081 | $0.905 \pm 0.010$ | -0.045 | $0.820 \pm 0.012$ | -0.045 |
| Feature only | $0.915 \pm 0.009$ | -0.045 | $0.660 \pm 0.014$ | -0.061 | $0.865 \pm 0.012$ | -0.085 | $0.760 \pm 0.015$ | -0.105 |
| w/ LayerNorm | $0.940 \pm 0.010$ | -0.020 | $0.695 \pm 0.012$ | -0.026 | $0.925 \pm 0.011$ | -0.025 | $0.835 \pm 0.013$ | -0.030 |
| Transformer Only | $0.875 \pm 0.013$ | -0.085 | $0.605 \pm 0.016$ | -0.116 | $0.845 \pm 0.014$ | -0.105 | $0.730 \pm 0.017$ | -0.135 |

Table 23: Ablation study results on CKD and MIMIC-IV datasets. Values denote mean $\pm$ 95% confidence intervals over 5 runs. $\Delta$ shows absolute performance drop from the full CoI model.

## E   CHAIN-OF-INFLUENCE VISUALIZATIONS

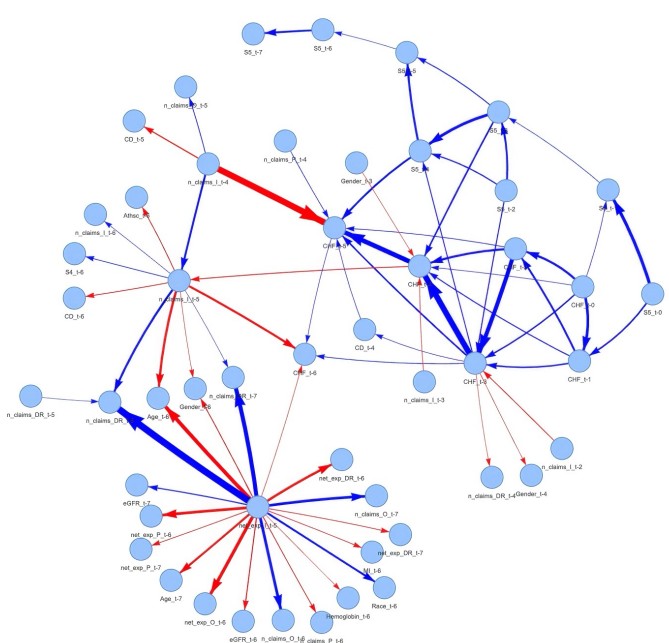

Figure 5: Chain-of-Influence network visualization for CKD progression prediction. Nodes represent clinical features at specific time points (e.g., eGFR_t-5 indicates eGFR at 5 time steps before prediction). Directed edges capture learned temporal dependencies between features, with red edges indicating risk-enhancing influences and blue edges representing protective relationships. The network reveals complex cascading pathways from early cardiovascular events through kidney function decline to healthcare utilization escalation, demonstrating CoI's ability to capture clinically meaningful temporal-feature interactions that drive disease progression.

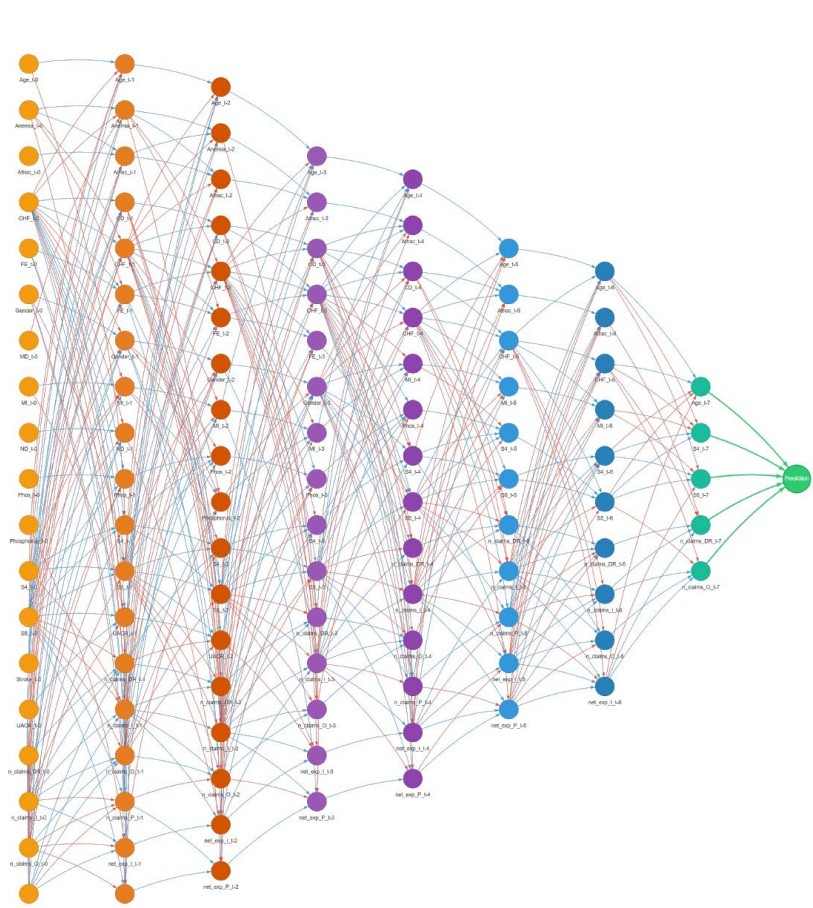

Figure 6: Top-5 patient-specific Chain-of-Influence graph for a single patient, showing the most influential feature–time pairs and their directed paths into the prediction node. Time Interval: $t_0 \rightarrow prediction$.

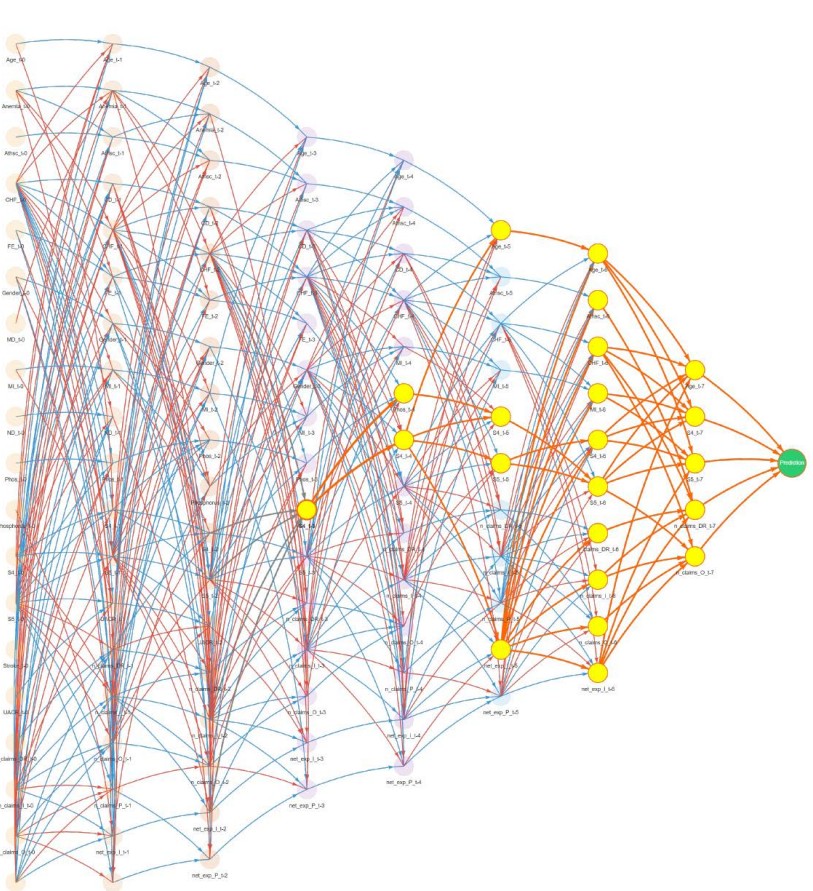

Figure 7: Path-centric view anchored at CKD Stage 4 at $t_3$, highlighting the high-influence chain through subsequent visits that drives the final ESRD risk.

