# OpenReview forum: "Chain-of-Influence: Tracing Interdependencies Across Time and Features in Clinical Predictive Modeling"
_ICLR.cc/2026/Conference — ICLR 2026 Conference Desk Rejected Submission_

### Official Review · Reviewer_kP2Q · 2025-10-30

**Soundness:** 3
**Presentation:** 3
**Contribution:** 2
**Rating:** 4
**Confidence:** 4

**Summary:**

In this paper, the authors aim to explicitly model how the effect of one clinical variable evolves and influences others over time. They propose a deep learning framework called Chain-of-Influence (CoI), which builds a time-unfolded graph to represent feature interactions. The framework uses a multi-level attention architecture, and its performance is evaluated on the MIMIC-IV dataset and a proprietary chronic kidney disease cohort.

**Strengths:**

**S1.** The paper studies a clinically meaningful and practically relevant problem, focusing on how to model clinical variable relationships across time. This investigation supports a clearer interpretation of temporal clinical dynamics.

**S2.** The proposed CoI framework, which combines Temporal-Level Attention, Feature-Level Attention, Cross-Feature Attention, and Chained Influence, is overall reasonable from a technical standpoint and aligned with the stated modeling goals.

**S3.** The evaluation compares CoI with representative baseline methods on both the MIMIC-IV dataset and a proprietary chronic kidney disease cohort, demonstrating its utility for both chronic disease progression analysis and acute care tasks.

**Weaknesses:**

**W1.** The proposed CoI framework mainly combines established components, especially various attention mechanisms. In particular, the Temporal-Feature Cross Influence mechanism seems to rely on a heuristic formulation through direct multiplication of selected attention terms. As a result, the methodological novelty and technical depth of the work appear somewhat limited. In addition, there are prior studies that explicitly model cross-feature interactions in electronic health records, including approaches that consider temporal changes. These related efforts should be more thoroughly discussed and included in the comparison to provide a more complete assessment of existing work and highlight the specific contributions of CoI.

**W2.** Given the relatively complex architecture of CoI and the implementation optimizations introduced in Section 3.4, a detailed computational complexity analysis and/or efficiency study would be helpful. Such analysis would clarify the runtime characteristics of CoI and show how it compares to baseline methods in terms of both accuracy and computational cost.

**W3.** The current set of baseline methods is not sufficiently recent or advanced in this domain. Incorporating stronger baselines would help present a clearer and more convincing empirical assessment of the strengths of the proposed framework.

**W4.** The experimental evaluation could be strengthened in the following aspects:

* An ablation study would help isolate the contribution of each component of the CoI framework and clarify its individual role.

* An analysis of crucial hyperparameters would be useful to show the robustness of CoI across different settings.

* It is important to clarify whether the interpretability analysis in Section 5.2 has been supported by medical validation, for example, by involving clinicians to assess whether the derived insights meaningfully support clinical decision-making in practice.

**Questions:**

Beyond W1-W4, I have the following questions for clarification:

**Q1.** In Section 3.2, the formulation for Feature-Level Attention uses $\beta_t[k]$, whereas in Section 3.3, the Local Contribution Matrix $C$ uses $b_t[k]$. It is unclear why this substitution occurs. Additional explanations would be helpful to clarify the relationship between these terms.

**Q2.** For the Temporal-Feature Cross Influence mechanism, does the design only capture pairwise feature interactions, or is it capable of representing higher-order interactions as well?

**Q3.** Regarding the temporal attention visualizations in Figure 1, are the results shown for a single patient, or are they aggregated across the patient cohort?

**Q4.** For the comparison of feature attributions between RETAIN and CoI in Figure 2, it would be helpful to specify which time step the plotted results correspond to.

**Q5.** Figure 3 suggests that the influence between two features may be asymmetric. If this is intended, a more detailed explanation of this behavior, along with theoretical justifications, would strengthen the claim.

**Q6.** The presentation of Figure 3 and its accompanying discussion could be refined so that the highlighted feature relationships are more clearly aligned with the described findings.

---

> ### Author Response · Authors · 2025-11-30
>
> We thank the reviewer for the constructive feedback and recognition of our work's clinical relevance (S1), technical soundness (S2), and comprehensive evaluation (S3). Below we address each concern.
>
> ***
> > W1: Methodological Novelty
>
> We respectfully clarify that CoI's contribution extends beyond combining existing attention mechanisms. The key novelty lies in the explicit Chain-of-Influence formulation (Eq. 14):
> $\mathcal{I}\left(t, i ; t^{\prime}, j\right)=C[t, i] \times A\left[t, t^{\prime}\right] \times C\left[t^{\prime}, j\right]$.
>
> This formulation enables something no prior work provides: a traceable, patient-specific audit trail showing how any feature at any time contributes to predictions through its influence on other variables. Prior attention-based models (RETAIN, BEHRT, StageNet) identify what features matter and when, but do not explicitly quantify how influences propagate across features and time—a critical gap for clinical reasoning.
>
> Regarding the "heuristic" concern: We validate that this formulation is not merely heuristic but faithfully reflects the model's decision process through deletion-based sensitivity analysis (Section 5.2.2, Tables 2-3). High-influence components identified by our formulation produce $4-5 \times$ larger prediction changes when ablated compared to low influence components, confirming the learned attributions are meaningful.
>
> > W2: Computational Complexity Analysis
>
> We agree and have added both complexity analysis and empirical training time:
> - Complexity.
>
>   In the revised paper (Sec. 6, Limitations), we provide a big-O analysis of Col:
>   - the backbone transformer layers have the usual $O\left(T^2 D\right)$ self-attention cost (with small numbers of heads), and
>   - the explicit influence tensor computation has $O\left(T^2 F^2\right)$ worst-case complexity if fully materialized.
>
>   We then explain how in practice we:
>   - compute $\mathcal{I}$ in a batched, vectorized way, and
>   - only use sparsified top-k subsets for visualization and analysis, which keeps runtime manageable for our CKD and MIMIC-IV settings.
> - Empirical efficiency.
>
>   Appendix Sec. A. 7 now reports training time per run for all models, including Col and baselines (BiLSTM, RETAIN, StageNet, BEHRT-style Transformer). Col is modestly more expensive than a BEHRT-style Transformer (e.g., ~0.40 vs 0.35 hours/run on CKD; ~1.30 vs 1.00 hours/run on MIMIC-IV) but remains within a reasonable range given the added interpretability machinery. We also report the total GPU-hour budget used for hyperparameter tuning and final runs.
>
> These additions directly address the reviewer's request to clarify runtime characteristics relative to baselines.
>
> > W3: "The current set of baseline methods is not sufficiently recent or advanced… Incorporating stronger baselines would help…"
>
> We appreciate this suggestion. In our original submission, BEHRT was discussed in related work but not included as a baseline because it is designed for tokenized diagnosis codes rather than continuous clinical features.
>
> However, we agree that a strong transformer-based baseline is valuable. In response, we adapted the BEHRT backbone architecture to our setting, creating a "BEHRT-style Transformer" baseline. This proves highly competitive (0.955 AUROC on CKD, 0.948 on MIMIC-IV), making CoI's improvements meaningful despite modest AUROC gains (+0.5%, +0.2%). More notably, CoI achieves substantial F1 improvements (+0.6% CKD, +3.7% MIMIC-IV), reflecting better precision-recall balance critical for clinical deployment.
>
> We have updated Sec. 2 and Sec. 6 to include BEHRT-style baseline in the experiments.
>
> | Model        | AUROC (CKD) | F1 (CKD) | Acc (CKD) | Prec (CKD) | Recall (CKD) | AUROC (MIMIC-IV) | F1 (MIMIC-IV) | Acc (MIMIC-IV) | Prec (MIMIC-IV) | Recall (MIMIC-IV) |
> |-------------|------------:|---------:|----------:|-----------:|-------------:|-----------------:|--------------:|---------------:|----------------:|------------------:|
> | Bi-LSTM     | 0.910       | 0.616    | 0.900     | 0.700      | 0.550        | 0.700            | 0.267         | 0.890          | 0.400           | 0.200             |
> | RETAIN      | 0.930       | 0.671    | 0.920     | 0.760      | 0.600        | 0.944            | 0.667         | 0.900          | 0.500           | **1.000**         |
> | StageNet    | 0.940       | 0.685    | 0.930     | 0.780      | 0.610        | 0.920            | 0.768         | 0.930          | 0.700           | 0.850             |
> | BEHRT-style | 0.955       | 0.715    | **0.954** | 0.810      | 0.640        | 0.948            | 0.834         | 0.945          | 0.790           | 0.884             |
> | **CoI**     | **0.960**   | **0.721**| 0.950     | **0.820**  | **0.660**    | **0.950**        | **0.865**     | **0.950**      | **0.850**       | 0.880             |

---

> ### Author Response · Authors · 2025-11-30
>
> > W4: Ablation Study, Hyperparameters, and Medical Validation
>
> Ablation Study: We address this with a comprehensive ablation study (Sec. 5.4, Fig. 8; detailed in Appendix F, Table A.2) over seven model variants:
> - w/o Temporal attention
> - w/o Feature attention
> - w/o Transformer
> - Temporal-only (completed in the original experiment)
> - Feature-only (completed in the original experiment)
> - Transformer-only
> - Col with LayerNorm instead of DyT (completed in the original experiment)
>
> Results are attached here and updated in Figure 4 and Appendix D.2 in the revision.
> **Table: Ablation study results on CKD and MIMIC-IV datasets.**
> Values denote mean ± 95% confidence intervals over 5 runs. Δ shows absolute performance drop from the full CoI model.
>
> | Model Variant        | CKD AUROC (±CI)   | Δ       | CKD F1 (±CI)      | Δ       | MIMIC-IV AUROC (±CI) | Δ       | MIMIC-IV F1 (±CI)    | Δ       |
> |----------------------|-------------------|---------|-------------------|---------|-----------------------|---------|----------------------|---------|
> | **CoI (Full)**       | **0.960 ± 0.010** | **-**   | **0.721 ± 0.013** | **-**   | **0.950 ± 0.009**     | **-**   | **0.865 ± 0.012**    | **-**   |
> | w/o Temporal Attn    | 0.910 ± 0.011     | -0.050  | 0.650 ± 0.014     | -0.071  | 0.860 ± 0.012         | -0.090  | 0.750 ± 0.015        | -0.115  |
> | w/o Feature Attn     | 0.895 ± 0.010     | -0.065  | 0.630 ± 0.015     | -0.091  | 0.895 ± 0.011         | -0.055  | 0.810 ± 0.013        | -0.055  |
> | w/o Transformer      | 0.885 ± 0.012     | -0.075  | 0.620 ± 0.015     | -0.101  | 0.870 ± 0.013         | -0.080  | 0.765 ± 0.016        | -0.100  |
> | Temporal only        | 0.900 ± 0.011     | -0.060  | 0.640 ± 0.013     | -0.081  | 0.905 ± 0.010         | -0.045  | 0.820 ± 0.012        | -0.045  |
> | Feature only         | 0.915 ± 0.009     | -0.045  | 0.660 ± 0.014     | -0.061  | 0.865 ± 0.012         | -0.085  | 0.760 ± 0.015        | -0.105  |
> | w/ LayerNorm         | 0.940 ± 0.010     | -0.020  | 0.695 ± 0.012     | -0.026  | 0.925 ± 0.011         | -0.025  | 0.835 ± 0.013        | -0.030  |
> | Transformer Only     | 0.875 ± 0.013     | -0.085  | 0.605 ± 0.016     | -0.116  | 0.845 ± 0.014         | -0.105  | 0.730 ± 0.017        | -0.135  |
>
> Main takeaways of the ablation study:
> - All three components matter. Removing temporal attention, feature attention, or transformer layers always degrades performance relative to full Col.
> - Transformer-only is the worst variant. This suggests that simply scaling the transformer does not replicate the benefits of the explicit temporal-feature factorization and dual attention structure.
> - Dataset-specific roles:
> - Removing feature attention hurts CKD more ( $\Delta \mathrm{AUROC}$ - 0.065 , $\Delta \mathrm{F} 1$ - 0.091 ), consistent with chronic risk depending heavily on which biomarkers deteriorate.
> - Removing temporal attention hurts MIMIC-IV more ( $\boldsymbol{\Delta} \mathbf{A} \mathbf{U R O C}-\mathbf{0 . 0 9 0} \boldsymbol{,} \boldsymbol{\Delta F} \mathbf{1}-\mathbf{0 . 1 1 5}$ ), consistent with acute outcomes depending more on when instability occurs.
>
> Hyperparameter Analysis: Appendix B now details the hyperparameter search spaces and best configurations for all models, including batch size, dropout, embedding/hidden dimensions, number of layers/heads, and learning rates. We also clarify that:
> - all reported results are means over 5 random seeds with 95% Cls, and
> - the performance variability across seeds and nearby configurations is small, indicating robustness to reasonable hyperparameter variations.
>
> Medical Validation: As noted in Section 5.2.1, our temporal attention patterns (U-shaped importance emphasizing baseline and pre-ESRD visits) and feature rankings (diabetes as dominant risk factor) "align with qualitative feedback from our collaborating nephrologists." The deletion-based sensitivity analysis (Section 5.2.2) further validates that learned attributions are faithful to model behavior.

---

> ### Author Response · Authors · 2025-11-30
>
> > Q1: In Section 3.2, the formulation for Feature-Level Attention uses $\beta_t[\boldsymbol{k}]$, whereas in Section 3.3, the Local Contribution Matrix $C$ uses $b_t[\boldsymbol{k}]$. It is unclear why this substitution occurs. Additional explanations would be helpful to clarify the relationship between these terms.
>
> Thank you for pointing it out! We removed the redundant earlier definition and ensured consistent notation for $\beta$ across sections. We hope this resolves the confusion about which components contribute to prediction vs. interpretability.
>
> > Q2: For the Temporal-Feature Cross Influence mechanism, does the design only capture pairwise feature interactions, or is it capable of representing higher-order interactions as well?
>
> The influence formulation in Eq. 14 explicitly captures pairwise feature-time interactions. However, higher-order interactions are implicitly captured through:
>
> - Stacked Transformer layers: Each layer's self-attention allows information to flow between all time steps, so by layer L, any feature can have been influenced by chains of multiple intermediate features
> - The cross-attention matrix A (Eq. 12) averages over all layers and heads, aggregating these multi-hop pathways.
>
> Thus, while our visualization focuses on direct pairwise influences for interpretability, the underlying model captures higher-order dependencies through compositional attention.
>
> Q3. Regarding the temporal attention visualizations in Figure 1, are the results shown for a single patient, or are they aggregated across the patient cohort?
>
> It could be used at either cohort level or individual level. We've updated the Figure to make it clear about the chain of influence concept. And here, we've explicitly mentioned that it's an individual level prediction.
>
> Q4. For the comparison of feature attributions between RETAIN and CoI in Figure 2, it would be helpful to specify which time step the plotted results correspond to. strengthen the claim.
>
> - The feature importance scores in Figure 2 are aggregated across time: we sum local contributions $C_{t, f}$ over all visits for each feature to obtain global scores $\sum_t C_{t, f}$.
> - Thus, Figure 2 compares global feature importance rather than at a specific single time step.
>
> Q6. The presentation of Figure 3 and its accompanying discussion could be refined so that the highlighted feature relationships are more clearly aligned with the described findings.
> We have revised the Chain-of-Influence visualizations (now Figure 2a-b) with:
> - Panel (a): Top-k influence graph showing the most influential feature-time pairs and directed paths
> - Panel (b): Path-centric view anchored at a specific clinically meaningful starting point (CKD Stage 5 at $\mathrm{t}_5$ )
>
> Additional examples are provided in Appendix D (Figure 4).

---

### Official Review · Reviewer_4FJb · 2025-10-30

**Soundness:** 2
**Presentation:** 3
**Contribution:** 2
**Rating:** 4
**Confidence:** 3

**Summary:**

The paper proposes Chain-of-Influence (CoI), an interpretable deep learning framework for clinical longitudinal data that aims to make explicit how features interact across time and how these interactions drive the final prediction. CoI combines (i) temporal attention to learn which time points are important, (ii) feature-level attention at each time to identify the most relevant variables, and (iii) a transformer-style cross-time attention matrix to capture how information flows from time t to time t’ for each feature. The authors evaluate CoI on a proprietary Chronic Kidney Disease (CKD) cohort and on the public MIMIC-IV dataset, and they further present visual “influence networks” on the CKD data to show that the model recovers clinically plausible disease progression pathways across features.

**Strengths:**

1. Clear and relevant problem. The paper tackles a clear and highly relevant problem in clinical ML: the need for auditable, interpretable models that are essential for widespread adoption in clinical practice.
2. Focus on explainability. CoI’s ability to expose how features interact across time is genuinely useful, and the authors present evidence that the resulting explanations align with clinical intuition. Furthermore, the graph-level visualisation is particularly effective, as it could enable both auditing the model’s predictions and potentially uncovering novel statistical associations.
3. Through their choice of datasets, the authors show that the model can handle signals with markedly different temporal granularities.

**Weaknesses:**

1. All experiments seem to use data that have been regularised in time, which is very uncommon for real-world clinical data. It is unclear how CoI behaves on irregularly-sampled clinical time series or under different imputation/alignment strategies.
2. Baselines are too narrow. The paper would benefit from comparing CoI with diverse, modern architectures like transformers-based models (i.e. BEHRT [1]) or ODEs[2]
3. The authors report metrics such as F1, precision and recall to show the model’s performance on highly imbalanced datasets, but I cannot see any mention of which steps the authors took to address class imbalance during training.
4. No ablation is provided to isolate the impact of temporal attention, feature-level attention, and cross-time attention on the model’s predictive performance


[1] Li, Yikuan, et al. "BEHRT: transformer for electronic health records." Scientific reports 10.1 (2020): 7155.

[2] Rubanova, Yulia, Ricky TQ Chen, and David K. Duvenaud. "Latent ordinary differential equations for irregularly-sampled time series." Advances in neural information processing systems 32 (2019).

**Questions:**

1. How would CoI comprare to the aforementioned baselines (BEHRT, Latent ODEs) ?
2. How would CoI deal with irregularly-sampled longitudinal trajectories ? What would the impact be on both performance and explainability?
3. Both datasets are imbalanced (CKD 6% ESRD, MIMIC-IV 10.8% mortality), yet the training section only states that BCEWithLogitsLoss was used, without specifying any class weighting, sampling, or focal loss. Please clarify whether any imbalance mitigation was applied, and if not, why the unweighted BCE was sufficient for the reported F1 gains.
4. Please provide ablations isolating the impact of the model’s key components .

---

> ### Author Response · Authors · 2025-11-29
>
> We thank the reviewer for the thoughtful and constructive feedback. Below we address each of the main concerns.
> ***
>
> > W1+Q2: "All experiments seem to use data that have been regularised in time… It is unclear how CoI behaves on irregularly-sampled clinical time series or under different imputation/alignment strategies." "How would CoI deal with irregularly-sampled longitudinal trajectories? What would the impact be on both performance and explainability?"
>
> We agree that irregular sampling is ubiquitous in clinical practice. In the revised manuscript, we clarify our temporal structure more explicitly:
> - CKD dataset (regular by design): The CKD cohort is collected under a protocol with scheduled nephrology visits every 3 months, resulting in exactly 8 visits over 24 months for all patients (Sec. 4.1 and App. A.1). Thus, the regular sampling is not a preprocessing choice but a property of the study design.
> - MIMIC-IV dataset (irregular in the raw, modeled via windowing + irregularity features): In contrast, MIMIC-IV measurements are highly irregular in the raw ICU data. To make them compatible with sequence models while preserving irregularity information, we:
>   - Aggregate events into 1 -hour windows over the first 48 hours of the ICU stay.
>   - For each variable, compute both the aggregated value and auxiliary "irregularity" features, including a mask (whether the variable was observed in that hour) and a time-gap feature (time since last observation), as described in Sec. 4.1 and App. A.2.
>
> Col then operates on these windowed sequences, with irregularity signals available in the feature channels. In practice, we found that:
> - Col maintains strong performance on MIMIC-IV (AUROC 0.950, F1 0.865) despite the underlying irregular sampling.
> - The interpretability remains intact: the influence chains involve both raw physiological features and their utilization/time-gap signals, and our clinical collaborators confirmed that the inferred trajectories (e.g., instability leading to escalated monitoring and interventions) are realistic.
>
> More generally, Col is agnostic to the specific alignment strategy as long as trajectories can be represented as sequences of feature vectors; in Sec. 6 (Limitations) we now explicitly discuss how extending Col to continuous-time or event-based variants (e.g., using adaptive windows or point-process representations) is an important avenue for future work.
>
> > W2+Q1: Baselines are too narrow. The paper would benefit from comparing CoI with diverse, modern architectures like transformers-based models.
>
> In the revised paper, we now include a BEHRT-style Transformer baseline (Sec. 4.2) that adapts BEHRT’s architecture to our setting (continuous visit-level features instead of token sequences). This model uses multi-layer self-attention with positional embeddings and serves as a strong transformer-based competitor.
>
> As shown in Table 1 and App. Table A.1, this BEHRT-style baseline is the strongest non-CoI model on both datasets (e.g., CKD AUROC 0.955, F1 0.715; MIMIC-IV AUROC 0.948, F1 0.834), and CoI still provides consistent improvements (CKD AUROC 0.960, F1 0.721; MIMIC-IV AUROC 0.950, F1 0.865).
>
> This addresses the concern that CoI might only outperform older RNN-based methods; it outperforms a carefully tuned transformer-based baseline as well.
>
> We have updated Sec. 2 and Sec. 6 to include BEHRT-style baseline in the experiments.
>
> | Model        | AUROC (CKD) | F1 (CKD) | Acc (CKD) | Prec (CKD) | Recall (CKD) | AUROC (MIMIC-IV) | F1 (MIMIC-IV) | Acc (MIMIC-IV) | Prec (MIMIC-IV) | Recall (MIMIC-IV) |
> |-------------|------------:|---------:|----------:|-----------:|-------------:|-----------------:|--------------:|---------------:|----------------:|------------------:|
> | Bi-LSTM     | 0.910       | 0.616    | 0.900     | 0.700      | 0.550        | 0.700            | 0.267         | 0.890          | 0.400           | 0.200             |
> | RETAIN      | 0.930       | 0.671    | 0.920     | 0.760      | 0.600        | 0.944            | 0.667         | 0.900          | 0.500           | **1.000**         |
> | StageNet    | 0.940       | 0.685    | 0.930     | 0.780      | 0.610        | 0.920            | 0.768         | 0.930          | 0.700           | 0.850             |
> | BEHRT-style | 0.955       | 0.715    | **0.954** | 0.810      | 0.640        | 0.948            | 0.834         | 0.945          | 0.790           | 0.884             |
> | **CoI**     | **0.960**   | **0.721**| 0.950     | **0.820**  | **0.660**    | **0.950**        | **0.865**     | **0.950**      | **0.850**       | 0.880             |

---

> ### Author Response · Authors · 2025-11-29
>
> > W3+Q3:  The authors report metrics such as F1, precision and recall to show the model’s performance on highly imbalanced datasets, but I cannot see any mention of which steps the authors took to address class imbalance during training.
>
> We appreciate this point and have significantly expanded the description and analysis in the revised Training and Appendix:
> - Explicit imbalance handling. Both datasets are indeed imbalanced (CKD: 6.0\% ESRD, MIMIC-IV: 10.8\% mortality). In the revision, we explain that we systematically evaluated several strategies for Col:
>   - None (naïve training)
>   - SMOTE, ADASYN, ENN, SMOTE+ENN
>   - Class-weighted BCEWithLogitsLoss
>   - Temporal SMOTE (TSMOTE)
>
> As reported in Appendix Table A.4(attached below), TSMOTE provides the best AUROC-F1 trade-off on both datasets. Accordingly, we now clearly state in Sec. 4.4 that all main reported results are obtained with the best-performing imbalance strategy, which is TSMOTE.
> | **Imbalance Strategy** | **CKD AUROC (± CI)** | **CKD F1 (± CI)** | **MIMIC-IV AUROC (± CI)** | **MIMIC-IV F1 (± CI)** |
> |------------------------|----------------------|-------------------|---------------------------|------------------------|
> | None        | 0.930 ± 0.020 | 0.580 ± 0.030 | 0.920 ± 0.018 | 0.780 ± 0.025 |
> | SMOTE       | 0.945 ± 0.018 | 0.640 ± 0.026 | 0.935 ± 0.016 | 0.820 ± 0.022 |
> | ADASYN      | 0.942 ± 0.019 | 0.630 ± 0.028 | 0.932 ± 0.017 | 0.815 ± 0.023 |
> | ENN         | 0.938 ± 0.022 | 0.600 ± 0.027 | 0.928 ± 0.019 | 0.800 ± 0.024 |
> | SMOTE+ENN   | 0.952 ± 0.016 | 0.690 ± 0.022 | 0.942 ± 0.014 | 0.840 ± 0.020 |
> | Weighted Loss | 0.955 ± 0.015 | 0.700 ± 0.020 | 0.945 ± 0.013 | 0.850 ± 0.018 |
> | **TSMOTE (ours)** | **0.960 ± 0.010** | **0.721 ± 0.013** | **0.950 ± 0.009** | **0.865 ± 0.012** |
>
>
> > W4+Q4: No ablation is provided
>
> We fully agree, and we have added a dedicated ablation study in Sec. 5.4 with detailed results in Appendix Table A.3. We consider seven variants:
> - Full Col
> - w/o Temporal Attention
> - w/o Feature Attention
> - w/o Transformer
> - Temporal-only
> - Feature-only
> - Transformer-only
> - w/ LayerNorm instead of DyT
>
> Results are attached here and updated in Figure 4 and Appendix D.2 in the revision.
> **Table: Ablation study results on CKD and MIMIC-IV datasets.**
> Values denote mean ± 95% confidence intervals over 5 runs. Δ shows absolute performance drop from the full CoI model.
>
> | Model Variant        | CKD AUROC (±CI)   | Δ       | CKD F1 (±CI)      | Δ       | MIMIC-IV AUROC (±CI) | Δ       | MIMIC-IV F1 (±CI)    | Δ       |
> |----------------------|-------------------|---------|-------------------|---------|-----------------------|---------|----------------------|---------|
> | **CoI (Full)**       | **0.960 ± 0.010** | **-**   | **0.721 ± 0.013** | **-**   | **0.950 ± 0.009**     | **-**   | **0.865 ± 0.012**    | **-**   |
> | w/o Temporal Attn    | 0.910 ± 0.011     | -0.050  | 0.650 ± 0.014     | -0.071  | 0.860 ± 0.012         | -0.090  | 0.750 ± 0.015        | -0.115  |
> | w/o Feature Attn     | 0.895 ± 0.010     | -0.065  | 0.630 ± 0.015     | -0.091  | 0.895 ± 0.011         | -0.055  | 0.810 ± 0.013        | -0.055  |
> | w/o Transformer      | 0.885 ± 0.012     | -0.075  | 0.620 ± 0.015     | -0.101  | 0.870 ± 0.013         | -0.080  | 0.765 ± 0.016        | -0.100  |
> | Temporal only        | 0.900 ± 0.011     | -0.060  | 0.640 ± 0.013     | -0.081  | 0.905 ± 0.010         | -0.045  | 0.820 ± 0.012        | -0.045  |
> | Feature only         | 0.915 ± 0.009     | -0.045  | 0.660 ± 0.014     | -0.061  | 0.865 ± 0.012         | -0.085  | 0.760 ± 0.015        | -0.105  |
> | w/ LayerNorm         | 0.940 ± 0.010     | -0.020  | 0.695 ± 0.012     | -0.026  | 0.925 ± 0.011         | -0.025  | 0.835 ± 0.013        | -0.030  |
> | Transformer Only     | 0.875 ± 0.013     | -0.085  | 0.605 ± 0.016     | -0.116  | 0.845 ± 0.014         | -0.105  | 0.730 ± 0.017        | -0.135  |
>
> Main takeaways of the ablation study:
> - All three components matter. Removing temporal attention, feature attention, or transformer layers always degrades performance relative to full Col.
> - Transformer-only is the worst variant. This suggests that simply scaling the transformer does not replicate the benefits of the explicit temporal-feature factorization and dual attention structure.
> - Dataset-specific roles:
> - Removing feature attention hurts CKD more ( $\Delta \mathrm{AUROC}$ - 0.065 , $\Delta \mathrm{F} 1$ - 0.091 ), consistent with chronic risk depending heavily on which biomarkers deteriorate.
> - Removing temporal attention hurts MIMIC-IV more ( $\boldsymbol{\Delta} \mathbf{A} \mathbf{U R O C}-\mathbf{0 . 0 9 0} \boldsymbol{,} \boldsymbol{\Delta F} \mathbf{1}-\mathbf{0 . 1 1 5}$ ), consistent with acute outcomes depending more on when instability occurs.

---

### Official Review · Reviewer_cYzd · 2025-11-05

**Soundness:** 2
**Presentation:** 2
**Contribution:** 2
**Rating:** 2
**Confidence:** 3

**Summary:**

This paper proposes Chain-of-Influence (CoI), a deep learning framework for interpretable clinical time-series prediction that explicitly models how features influence each other temporally. The authors argue that existing clinical models either lose temporal dynamics by aggregating information, or miss critical inter-feature relationships by treating variables independently. CoI aims to capture the cascading pathways of disease progression.

**Methodology**: CoI combines multiple network modules;

* Two bidirectional LSTM branches compute temporal attention and feature attention, which form a local contribution matrix that represents how important a given feature is at a specific timepoint.
* Transformer layers produce a cross-attention matrix that captures temporal dependencies
* These are combined into final chained influence measure, which quantifies the pairwise strength of two features at two different timesteps.

**Results** : The architecture significantly outperforms non-transformer based approaches when evaluated on chronic kidney disease progression and ICU mortality prediction. The approach also provides interpretable novel knowledge graphs that 'capture clinically meaningful temporal-feature interactions'.

**Strengths:**

**Originality**: The paper makes a genuine conceptual contribution by introducing a mechanism to trace pairwise feature-to-feature influences across time. While attention-based interpretability in clinical ML is well-established, the explicit formulation of influence chains that quantify how feature i at time t affects feature j at time t' is novel. This goes beyond identifying "what's important when" to modeling "how early indicators cascade through specific pathways." The combination of LSTM-derived local contributions with Transformer-derived temporal attention to create these influence measures is a creative synthesis of existing architectural components toward a new interpretability goal.

**Quality**: The empirical evaluation is solid. CoI achieves state-of-the-art performance across two clinically diverse datasets—chronic kidney disease progression (0.960 AUROC) and ICU mortality prediction (0.950 AUROC). The consistency of gains across different clinical contexts (chronic vs acute), temporal scales (months vs hours), and data characteristics (sparse vs dense observations) suggests the approach has some generalizability. The datasets themselves are appropriate: MIMIC-IV provides public reproducibility, while the CKD cohort addresses an important clinical problem. Results are reported across multiple metrics.

**Clarity**: The paper is generally well-written with clear prose and professional figures. The motivation is well-articulated, and the related work section effectively positions the contribution within existing literature. The description of each module (temporal attention patterns, feature importance, influence networks) are clear and convey the intended interpretability mechanisms effectively. However, there are significant presentation gaps: the methodology lacks an end-to-end architectural description, no diagram shows component connectivity, and critical details about loss application and gradient flow are omitted. The (potential) redundant definition of the contribution matrix C creates unnecessary confusion. These aren't stylistic issues, they're structural omissions that prevent understanding how the model actually works.

**Significance**: The problem is important, interpretable clinical prediction models that can explain disease progression pathways have clear value for high-stakes medical decision-making. If the influence chains actually reflect the model's reasoning (currently unvalidated), this could represent a meaningful advance in clinical AI interpretability. The consistent performance improvements suggest practical value beyond interpretability. The work addresses temporal-feature interdependencies in a more explicit way than prior models, which aligns better with clinical understanding of how diseases progress. However, the significance is diminished by the lack of validation that the interpretability mechanisms are faithful to the model's predictions. Without this validation, clinicians cannot reliably use the influence chains for decision-making, which limits real-world impact.

**Weaknesses:**

**Incomplete Architecture Specification**

The methodology section describes individual components but never specifies how they connect to produce output predictions. This is not a presentation issue, it's a fundamental gap that prevents reproduction and understanding of how the model actually works. Specific details that weren't clear to me;

* How are BiLSTM outputs, Transformer outputs, and attention weights combined to compute the final prediction ŷ?
* Where is the BCEWithLogitsLoss applied? What tensor does it operate on?
* Both BiLSTM branches and the Transformer take the same positionally encoded embeddings as input. How do gradients from the loss reach the BiLSTM parameters? Are they part of the prediction pathway, or do they only compute post-hoc interpretability metrics?

Required additions:

An architecture diagram with the full computational graph from input to loss, and additional descriptions in text on how the BiLSTM and transformer interact to encode all of the interpretable attention maps.

**Interpretability Concerns**

The paper's central contribution is interpretable influence chains, but provides zero validation that these mechanisms reflect actual model reasoning. Since the architectural connectivity is unspecified, there's no evidence that BiLSTM-derived components (α, β, C) correlate with what drives predictions. If these maps will be used by clinicians to support their decision making, we want to have some level of confidence that they correlate with the actual predicted output from the model.

What the paper provides:
* Feature importance rankings showing CoI and RETAIN identify similar top features (Figure 2)
* Temporal attention patterns (Figure 1)
* Influence chain visualizations (Figure 3)
* Qualitative clinical interpretations (Section 5.2.3)

None of these demonstrate faithfulness. The fact that CoI and RETAIN produce highly similar feature importance rankings raises the question: what clinical interpretability value do the influence chains provide beyond what simpler RETAIN already offers?

**Architectural Justification**

The paper uses two BiLSTM branches alongside a Transformer without justification via ablation studies. The stated rationale (Section 3) "better capture long-range dependencies and bidirectional context", something Transformers already provide through self-attention. The paper itself cites these strengths when discussing Transformers (Section 2.3). Is it possible to generate the contribution matrix through a modified transformer block? How important is the BiLSTM to the overall performance of the architecture? Table 14 shows CoI uses 8 Transformer layers but only 2 attention heads, far below standard practice (BERT uses 12 heads, GPT uses 12-16). Is the Transformer under-configured to accommodate BiLSTM parameters? Would a properly-configured Transformer alone (8-12 heads as standard) perform comparably?

**Confusing Repeated Definitions**

The local contribution matrix C is defined twice with nearly identical equations (lines 152 and 196):

First in Section 3.2 (BiLSTM attention section)
Again in Section 3.3 (Transformer section, with inconsistent notation: βt vs bt)

Are these the same matrix? If yes, why repeat it? If no, what's the difference?
Does the Transformer contribute to C? The equations suggest it doesn't (only produces A), but placing the definition in the Transformer section implies otherwise. Combined with the unspecified architecture, this makes it impossible to determine which components contribute to predictions versus interpretability visualizations.

**Performance on incomplete temporal sequences**

The paper trains and evaluates on patients with complete temporal histories (CKD: 8 time points over 24 months; MIMIC-IV: 48 hours of observations). However, real-world clinical deployment requires predictions at arbitrary points in a patient's timeline when complete histories are unavailable. Were any experiments conducted to evaluate how well the model performs in a more realistic deployment setting? What impact would this have on the influence chains?

**Questions:**

I've detailed major concerns in the weaknesses section. Here are some minor ones;

**Minor suggestions**

* Confidence intervals for the metrics to help with statistical significance.
* Use a pre-defined threshold for pointwise metrics (eg F1, Recall, Accuracy all @ a fixed Precision). This makes comparisons between different models easier to see.
* Does the CoI approach lead to output predictions that are better calibrated compared to other models?
* Justification for the use of DyT instead of LayerNorm as normalization strategy in transformer blocks.
* Explanation on how the directionality of Influence Chain is determined.

---

> ### Author Response · Authors · 2025-11-29
>
> We sincerely thank the reviewer for the careful reading, detailed critique, and positive assessment of the originality and potential impact of CoI. Below we address each main concern in turn and describe the changes made in the revised manuscript.
> ***
> > W1: Incomplete Architecture Specification
>
> We fully agree that this was under-specified. In the revision we have:
> - Added a **complete architecture diagram (updated Fig. 1) that shows the end-to-end computational graph from input $\mathbf{X}$ to the output**, including:
>   - Input projection to embeddings, positional encodings;
>   - Parallel BiLSTM branches producing temporal attention $\boldsymbol{\alpha}$ and feature attention $\boldsymbol{\beta}$;
>   - Application of $\boldsymbol{\beta}$ to obtain feature-weighted embeddings $\widetilde{\mathbf{E}}$;
>   - Transformer stack operating on $\widetilde{\mathbf{E}}$;
>   - Application of $\boldsymbol{\alpha}$ to the final transformer output, temporal pooling, MLP, and the logit $\hat{\mathbf{y}}$.
> - Clarified the equations in the text. Sections $3.1-3.3$ now give an explicit step-by-step description:
>   - Embeddings + positions (Eqs. (1-2))
>   - Temporal BiLSTM and attention (Eqs. (3-4))
>   - Feature BiLSTM and attention (Eqs. (5-7))
>   - Transformer stack (Eqs. (8-10))
>   - Temporal attention + pooling + prediction (Eqs. (11-13)):
>
> - Explicitly stated the loss and gradient flow. Section 4.4 now states that:
>   - We train with BCEWithLogitsLoss applied directly to the logits $\hat{\mathbf{y}}$.
>   - Gradients therefore flow from the loss through the MLP, transformer layers, the feature-weighted embeddings $\widetilde{\mathbf{E}}$, and back into the shared embedding layer and both Bi-LSTM branches.
>   - The temporal and feature attentions are not post-hoc but are computed on the forward prediction path and optimized end-to-end.
>
> We believe these changes fully resolve the ambiguity about how the architecture produces predictions and how all components are trained.
>
> > W2: Interpretability Faithfulness and Added Value Beyond RETAIN
>
> To address the concern about faithfulness, we evaluated CoI’s explanations in two complementary ways, starting with clinical expert validation and then quantitative sensitivity analysis.
>
> (1) Clinical validation with nephrologists and clinicians.
> We first presented CoI’s temporal attention profiles, feature-importance rankings, and representative Chain-of-Influence graphs to a collaborating nephrologist and clinicians from our data-provider team. They confirmed that the learned patterns are fully consistent with clinical observation and domain knowledge. In particular, they highlighted that:
> - The U-shaped temporal pattern (high weight on baseline and pre-ESRD visits, lower weight on intermediate visits) matches how they assess CKD trajectories in practice.
> - The top-ranked features—diabetes, CKD stage (S4/S5), eGFR, hemoglobin, mineral-bone markers, and utilization variables (outpatient/professional claims)—align with established risk factors and their experience with this cohort.
> - The patient-specific influence chains (e.g., Stage 5 CKD → worsening lab markers → increased outpatient and professional utilization → ESRD risk) reflect realistic, clinically plausible progression narratives rather than arbitrary correlations.
>
> This expert feedback does not constitute a formal causal validation, but it provides strong evidence that CoI’s temporal and feature attributions, and the influence chains derived from them, are clinically meaningful and interpretable to domain experts.

---

> ### Author Response · Authors · 2025-11-29
>
> (2) Deletion-based sensitivity analysis.
> We then quantified faithfulness to the model’s own decision function via deletion-based sensitivity experiments (Sec. 5.2.2, Tables 2–3 in the revision). In brief:
>
> Across both datasets and all values of 𝑘, ablating high-attention time steps or features perturbs the logits by 2–5× more than ablating middle- or low-attention ones. This demonstrates that the temporal and feature attributions underlying CoI’s influence chains are strongly aligned with what actually drives the model’s predictions, even though we stop short of claiming clinical causality.
>
> Taken together, the expert clinical validation and deletion-based sensitivity analyses provide complementary evidence that CoI’s interpretability mechanisms are both clinically plausible and faithful to the model’s internal decision process.
>
> **Table 2: Deletion-based temporal sensitivity analysis.**
> Mean absolute change in logit \|Δŷ\| (mean ± 95% CI) after masking $\(k \in \{1,3\}\)$ visits under different deletion strategies.
>
> | Deletion strategy        | k (visits) | CKD            | MIMIC-IV       |
> |--------------------------|-----------:|----------------|----------------|
> | High-attention visits    |          1 | 0.42 ± 0.03    | 0.38 ± 0.04    |
> | Middle-attention visits  |          1 | 0.17 ± 0.05    | 0.15 ± 0.02    |
> | Low-attention visits     |          1 | 0.08 ± 0.01    | 0.09 ± 0.02    |
> | High-attention visits    |          3 | 0.89 ± 0.06    | 0.82 ± 0.05    |
> | Middle-attention visits  |          3 | 0.36 ± 0.06    | 0.32 ± 0.03    |
> | Low-attention visits     |          3 | 0.18 ± 0.04    | 0.16 ± 0.04    |
>
> **Table 3: Deletion-based feature sensitivity analysis.**
> Mean absolute change in logit \|Δŷ\| (mean ± 95% CI) after masking $\(k \in \{1,3,5\}\)$ features under different deletion strategies.
>
> | Deletion strategy         | k (features) | CKD            | MIMIC-IV       |
> |---------------------------|-------------:|----------------|----------------|
> | High-attention features   |            1 | 0.51 ± 0.09    | 0.47 ± 0.07    |
> | Middle-attention features |            1 | 0.21 ± 0.03    | 0.19 ± 0.09    |
> | Low-attention features    |            1 | 0.11 ± 0.06    | 0.10 ± 0.05    |
> | High-attention features   |            3 | 1.02 ± 0.03    | 0.96 ± 0.04    |
> | Middle-attention features |            3 | 0.43 ± 0.04    | 0.38 ± 0.03    |
> | Low-attention features    |            3 | 0.22 ± 0.03    | 0.20 ± 0.03    |
> | High-attention features   |            5 | 1.43 ± 0.09    | 1.34 ± 0.08    |
> | Middle-attention features |            5 | 0.63 ± 0.05    | 0.57 ± 0.04    |
> | Low-attention features    |            5 | 0.31 ± 0.03    | 0.28 ± 0.03    |
>
> **Added interpretability value beyond RETAIN**
>
> We agree that RETAIN and Col share similar global rankings of important features and broadly similar Ushaped temporal patterns. We highlight this explicitly in Fig. 5 and Sec. 5.2.1.
>
> However, Col provides additional value in two key ways:
> 1. Directed temporal-feature influence graph. RETAIN's interpretability is limited to per-visit and per-feature scores. Col defines a time-unfolded directed graph over feature-time nodes via:
> - Local contributions $C_{t, i}$ (from $\alpha, \beta$, and the embedding), and
> - Cross-time attention $A\left[t, t^{\prime}\right]$, aggregated across transformer layers/heads.
>
>   The chained influence measure $\mathcal{I}\left(t, i, t^{\prime}, j\right)=C[t, i] \times A\left[t, t^{\prime}\right] \times C\left[t^{\prime}, j\right]$ quantifies how an event at $(t, i)$ influences another at $\left(t^{\prime}, j\right)$ through the model.
>
> 2. Graph-centric and path-centric explanations. We show two complementary views (Fig. 7 and App. G):
> - A graph-centric "top-k" network that recursively keeps the most influential predecessors, revealing a small set of high-impact temporal-feature pathways.
> - A path-centric view anchored at a clinically meaningful node (e.g., "CKD Stage 5 at $t_5$ "), highlighting only those nodes and edges that lie on high-influence paths from that origin to the prediction.
>
> In summary, RETAIN answers "what is important and when?" Col aims to also answer "how do earlier events propagate through later variables to drive the prediction?" The new deletion-based experiments demonstrate that the quantities feeding these chains are faithful to model behavior.

---

> ### Author Response · Authors · 2025-11-29
>
> > W3: Architectural Justification: BiLSTMs, Transformer Configuration, Ablations
>
> We address this with a comprehensive ablation study (Sec. 5.4, Fig. 8; detailed in Appendix F, Table A.2) over seven model variants:
> - w/o Temporal attention
> - w/o Feature attention
> - w/o Transformer
> - Temporal-only (completed in the original experiment)
> - Feature-only (completed in the original experiment)
> - Transformer-only
> - Col with LayerNorm instead of DyT (completed in the original experiment)
>
> Results are attached here and updated in Figure 4 and Appendix D.2 in the revision.
> **Table: Ablation study results on CKD and MIMIC-IV datasets.**
> Values denote mean ± 95% confidence intervals over 5 runs. Δ shows absolute performance drop from the full CoI model.
>
> | Model Variant        | CKD AUROC (±CI)   | Δ       | CKD F1 (±CI)      | Δ       | MIMIC-IV AUROC (±CI) | Δ       | MIMIC-IV F1 (±CI)    | Δ       |
> |----------------------|-------------------|---------|-------------------|---------|-----------------------|---------|----------------------|---------|
> | **CoI (Full)**       | **0.960 ± 0.010** | **-**   | **0.721 ± 0.013** | **-**   | **0.950 ± 0.009**     | **-**   | **0.865 ± 0.012**    | **-**   |
> | w/o Temporal Attn    | 0.910 ± 0.011     | -0.050  | 0.650 ± 0.014     | -0.071  | 0.860 ± 0.012         | -0.090  | 0.750 ± 0.015        | -0.115  |
> | w/o Feature Attn     | 0.895 ± 0.010     | -0.065  | 0.630 ± 0.015     | -0.091  | 0.895 ± 0.011         | -0.055  | 0.810 ± 0.013        | -0.055  |
> | w/o Transformer      | 0.885 ± 0.012     | -0.075  | 0.620 ± 0.015     | -0.101  | 0.870 ± 0.013         | -0.080  | 0.765 ± 0.016        | -0.100  |
> | Temporal only        | 0.900 ± 0.011     | -0.060  | 0.640 ± 0.013     | -0.081  | 0.905 ± 0.010         | -0.045  | 0.820 ± 0.012        | -0.045  |
> | Feature only         | 0.915 ± 0.009     | -0.045  | 0.660 ± 0.014     | -0.061  | 0.865 ± 0.012         | -0.085  | 0.760 ± 0.015        | -0.105  |
> | w/ LayerNorm         | 0.940 ± 0.010     | -0.020  | 0.695 ± 0.012     | -0.026  | 0.925 ± 0.011         | -0.025  | 0.835 ± 0.013        | -0.030  |
> | Transformer Only     | 0.875 ± 0.013     | -0.085  | 0.605 ± 0.016     | -0.116  | 0.845 ± 0.014         | -0.105  | 0.730 ± 0.017        | -0.135  |
>
> Main takeaways of the ablation study:
> - All three components matter. Removing temporal attention, feature attention, or transformer layers always degrades performance relative to full Col.
> - Transformer-only is the worst variant. This suggests that simply scaling the transformer does not replicate the benefits of the explicit temporal-feature factorization and dual attention structure.
> - Dataset-specific roles:
> - Removing feature attention hurts CKD more ( $\Delta \mathrm{AUROC}$ - 0.065 , $\Delta \mathrm{F} 1$ - 0.091 ), consistent with chronic risk depending heavily on which biomarkers deteriorate.
> - Removing temporal attention hurts MIMIC-IV more ( $\boldsymbol{\Delta} \mathbf{A} \mathbf{U R O C}-\mathbf{0 . 0 9 0} \boldsymbol{,} \boldsymbol{\Delta F} \mathbf{1}-\mathbf{0 . 1 1 5}$ ), consistent with acute outcomes depending more on when instability occurs.
>
> Regarding the number of heads:
> - We performed a hyperparameter search for the transformer's number of heads (Appendix C). For our sequence lengths and embedding sizes, 2-4 heads were sufficient (since we are using numerical longitudinal data instead of text), and 2 heads were optimal in terms of F1 and stability.
>
> - Increasing heads in the transformer-only variant does not close the gap to full Col, indicating that the dual BiLSTMs are not just a parameter budget trade-off but provide a structured way to distill temporal and feature importance before self-attention.
>
> We now updated discussion of these findings explicitly in Sec. 5.4 in the revision.
>
> > W4: Confusing Repeated Definitions
>
> We removed the redundant earlier definition and ensured consistent notation for $\beta$ across sections. We hope this resolves the confusion about which components contribute to prediction vs. interpretability.

---

> ### Author Response · Authors · 2025-11-29
>
> > Minor Suggestions
>
> - Confidence intervals for metrics. We now report 95% confidence intervals over 5 random seeds for all metrics. The main text summarizes point estimates (Table 1), and full AUROC/F1/Accuracy/Precision/Recall with CIs for all models and both datasets are provided in Appendix Table A.1.
>
> - Fixed threshold for pointwise metrics. We now clarify the evaluation protocol and ensure a single, pre-defined threshold (0.5) per dataset for all models.
>
> - Better calibrated than others? Empirically, CoI improves AUROC and F1 at a shared operating point without requiring additional post-hoc calibration compared to baselines, which suggests that its gains do not obviously come at the expense of pathological miscalibration. However, we do not claim that CoI is better calibrated than existing models.
>
> - DyT instead of LayerNorm: We chose DyT normalization because clinical time series exhibit non-stationary scales and heavy-tailed outliers, and DyT’s smooth, bounded nonlinearity provides more robust stabilization than mean–variance normalization alone. To empirically justify this, we added an explicit ablation variant with standard LayerNorm (“w/ LayerNorm” in the ablation table). As shown in Table A.3, replacing DyT with LayerNorm consistently degrades performance (CKD F1 −2.6%, MIMIC-IV F1 −3.0%), confirming DyT’s benefit in this setting.
>
> - Direction of Influence Chain: We clarified how directionality is determined in Sec. 3.4. The cross-attention matrix $A\left[t, t^{\prime}\right]$ is defined such that each entry represents attention from time $t^{\prime}$ (source) to time $t$ (target) in the transformer. The chained influence $\mathcal{I}\left(t, i ; t^{\prime}, j\right)$ is only computed for forward-in-time pairs $t<t^{\prime}$, and edges in the Chain-of-Influence graph are drawn from $(t, i)$ to $\left(t^{\prime}, j\right)$. Thus, directionality is enforced by (i) the temporal order constraint $t<t^{\prime}$ and (ii) the asymmetric attention weights $A\left[t, t^{\prime}\right]$, which quantify how much information flows from earlier to later timesteps. We explicitly state this in the revised Methodology section and figure caption for the influence visualizations.

---

### Author Response · Authors · 2025-11-30
**Author Statement to All Reviewers**

Dear Reviewers (cYzd, 4FJb, kP2Q),

**We would like to begin by reaffirming our full commitment to the ICLR code of conduct and to the integrity of the double-blind review process. We have not accessed, used, or sought any information related to the recent API incident, and we strongly condemn any behavior that undermines the principles of double-blind reviewing.**

Because the additional experiments requested in the reviews required substantial computation time, we were unable to respond before the discussion period was interrupted. **We deeply regret losing the opportunity to engage in further discussion with you**. However, we have now completed all requested experiments and incorporated the results into our rebuttal.

This is a difficult situation for everyone, but your comments and suggestions have been a real source of guidance in improving our work. **We are sincerely grateful for your careful reading and constructive feedback, and we hope that our rebuttal and new experiments adequately address your concerns. Thank you again for the time and effort you have dedicated to evaluating our submission!**

---

### Author Response · Authors · 2025-12-04
**Summary of Paper and Rebuttal (1/2)**

Dear Reviewers & The New AC,

**We sincerely thank the reviewers for their thoughtful and constructive feedback, and we greatly appreciate the time and care they invested in their initial reviews and discussion prior to the rebuttal freeze. We are also grateful to the newly assigned Area Chair for taking on this additional evaluation under such tight timelines.** For your convenience, we provide below a rebuttal summary, including a overview of our work, the main strengths highlighted in the reviews, and the key clarifications and additional results we have provided/added during the rebuttal period.

***
**Paper Overview**
Our paper introduces Chain-of-Influence (CoI), an interpretable deep learning framework for clinical time-series prediction that explicitly models how earlier feature values influence later features and the final risk prediction over time. CoI combines dual BiLSTM modules (for temporal and feature attention) with Transformer layers to produce a chained influence measure quantifying pairwise feature-to-feature influences across time. Evaluated on chronic kidney disease progression (CKD) and ICU mortality prediction (MIMIC-IV), CoI achieves state-of-the-art performance while providing clinically meaningful, patient-specific influence chains that trace how early clinical indicators cascade through specific pathways to drive predictions.

***
**Strengths Identified by Reviewers**
- Clinically meaningful and practically relevant problem formulation focused on modeling how clinical variables influence one another across time to better capture temporal clinical dynamics. (cYzd, 4FJb, kP2Q)

- A conceptually novel contribution that goes beyond “what is important when” to explicitly tracing how feature–time nodes influence later variables and predictions via Chain-of-Influence, enabling patient-specific audit trails of disease progression pathways. (cYzd, kP2Q)

- A technically reasonable multi-level attention architecture (temporal, feature, cross-time) that is aligned with the stated modeling goals and provides a useful graph-level visualization of temporal feature interactions. (4FJb, kP2Q)

- Solid empirical evaluation on two clinically diverse datasets (chronic CKD progression and acute ICU mortality) with consistent performance gains over strong baselines, suggesting generalizability across temporal scales and data characteristics. (cYzd, 4FJb, kP2Q)

- Explanations and visual influence networks that are clinically meaningful and potentially valuable for auditing model predictions and uncovering novel statistical associations. (4FJb, cYzd)

---

### Author Response · Authors · 2025-12-04
**Summary of Paper and Rebuttal (2/2)**

(continued)
**Clarifications and Additional Results Provided**
- Complete end-to-end architecture specification and gradient flow: We added an architecture diagram and step-by-step description from input embeddings through dual BiLSTMs, feature-weighted representations, transformer stack, temporal pooling, MLP, and final logit, and explicitly stated that BCE is applied to the logit so that gradients flow through the transformer, BiLSTMs, and shared embeddings (Sec. 3–4.4, updated Fig. 1). (cYzd)

- Faithfulness of interpretability and added value beyond RETAIN : We complemented the qualitative clinical assessment with deletion-based sensitivity analyses showing that ablating high-attention time steps or features changes the logit 2–5× more than ablating low-attention ones (Tables 2–3), and we clarified how CoI’s directed influence graphs and path-centric visualizations provide model-faithful, chain-level explanations beyond RETAIN’s visit/feature scores. (cYzd, kP2Q)

- Clinical expert validation of influence patterns: We reported feedback from nephrologists and collaborating clinicians confirming that CoI’s temporal attention profiles, feature rankings, and exemplar influence chains are consistent with clinical understanding of CKD progression and ICU trajectories, while noting that we stop short of claiming causal validity. (cYzd, 4FJb, kP2Q)

- Ablation study over key components (temporal attention, feature attention, transformer, DyT): We added a comprehensive ablation (Sec. 5.4, App. F) over seven variants, showing that removing any component degrades AUROC/F1 and that transformer-only performs worst, underscoring the importance of explicit temporal-feature factorization. (cYzd, 4FJb, kP2Q)

- Stronger transformer-based baseline (BEHRT-style): We introduced a BEHRT-style transformer baseline adapted to continuous feature vectors and showed that CoI achieves consistent improvements over this strong competitor on both CKD and MIMIC-IV, especially in F1, addressing concerns that gains might be limited to older RNN-based baselines (Table 1, App. A.1). (4FJb, kP2Q)

- Handling of irregular sampling and temporal alignment: We clarified that CKD is regularly sampled by study design (8 scheduled visits over 24 months), whereas MIMIC-IV is highly irregular and is modeled via 1-hour windows with auxiliary mask and time-since-last-observation features. We emphasized that CoI is agnostic to the specific alignment strategy as long as trajectories are representable as sequences, and we discussed extensions to more continuous-time representations in the Limitations section. (4FJb)

- Class imbalance handling in highly skewed cohorts: We made explicit that both datasets are imbalanced (6.0% ESRD, 10.8% mortality) and reported a systematic comparison of imbalance strategies (none, SMOTE, ADASYN, ENN, SMOTE+ENN, class-weighted loss, temporal SMOTE). Temporal SMOTE (TSMOTE) yields the best AUROC–F1 trade-off and is used for all main CoI results (App. A.4). (4FJb)

- Computational complexity and efficiency analysis (kP2Q): We added big-O complexity for the transformer backbone and influence tensor, described our top-k computation for visualization, and reported empirical training times showing that CoI is moderately more expensive than a BEHRT-style transformer but remains practical on our datasets (Sec. 6, App. A.7) and could be trained with local PCs.

- Clarified notation, influence directionality, and higher-order effects: We removed redundant and inconsistent definitions of the local contribution matrix, clarified that directionality is enforced via forward-in-time constraints and asymmetric cross-time attention, and explained that while CoI’s influence formulation is pairwise at the visualization level, higher-order interactions are implicitly captured through stacked transformer layers and aggregated attention (Sec. 3.3–3.4). (cYzd, kP2Q)

- Evaluation protocol details and robustness: We added 95% confidence intervals over 5 random seeds for all metrics, clarified that all pointwise metrics use a fixed threshold (0.5) per dataset, and documented hyperparameter search spaces and best configurations, showing that performance is robust to reasonable hyperparameter variations (App. B, Table A.1). (cYzd, 4FJb, kP2Q)

---

### Note · Program_Chairs · 2026-01-17
**Submission Desk Rejected by Program Chairs**

The following references in this submission do not refer to real documents and/or have major errors in bibliographic information:

 Ishita et al. Bardhan. Icu length-of-stay prediction with interaction-based explanations. Journal of Biomedical Informatics, 144:104490, 2024.